

# On Aethalometer measurement uncertainties and multiple scattering enhancement in the Arctic

John Backman[1], Lauren Schmeisser[2,*], Aki Virkkula[1], John A. Ogren[2,3], Eija Asmi[1], Sandra Starkweather[2,3], Sangeeta Sharma[4], Konstantinos Eleftheriadis[5], Taneil Uttal[3], Anne Jefferson[2], Michael Bergin[6], Alexander Makshtas[7]

[1]Finnish Meteorological Institute, Atmospheric Composition Research, Helsinki, Finland
[2]University of Colorado, Cooperative Institute for Research in Environmental Sciences, Boulder, USA
[3]National Oceanic and Atmospheric Administration, Earth System Research Laboratory, Boulder, USA
[4]Environment and Climate Change Canada, Climate Research Division, Downsview, Canada
[5]Institute of Nuclear and Radiological Science & Technology, Energy & Safety, Environmental Radioactivity Laboratory, NCSR "Demokritos", Athens, Greece
[6]Duke University, Civil and Environmental Engineering, Durham, USA
[7]Russian Federal Service for Hydrometeorology and Environmental Monitoring, Arctic and Antarctic Research Institute, St. Petersburg, Russia
[*]Now at University of Washington, Department of Atmospheric Sciences, Seattle, USA

*Correspondence to*: John Backman (john.backman@fmi.fi)

**Abstract.** Several types of filter-based instruments are used to estimate aerosol light absorption coefficients. Two significant results are presented based on Aethalometer measurements at six Arctic station from 2012–2014. First, an alternative method of post-processing the Aethalometer data is presented which reduces measurement noise and lowers the detection limit of the instrument more effectively than boxcar averaging. The biggest benefit of this approach can be achieved if instrument drift is minimized. Moreover, by using an attenuation threshold criterion for data post-processing, the relative uncertainty from the electronic noise the instrument is kept constant. This approach results in a time series with a variable collection time ($\Delta t$), but with a constant relative uncertainty with regard to electronic noise in the instrument. An additional advantage of this method is that the detection limit of the instrument will be lowered at small aerosol concentrations at the expense of temporal resolution, whereas there is little to no loss in temporal resolution at high aerosol concentrations (>2.1–6.7 Mm$^{-1}$ as measured by the Aethalometers). At high aerosol concentrations, minimizing the detection limit of the instrument is less critical. Second, utilizing co-located reference methods of aerosol absorption, a multiple scattering enhancement factor ($C_{ref}$) of 3.10 specific to low elevation Arctic stations is found. $C_{ref}$ is a fundamental part of most of the Aethalometer corrections available in literature, and this is the first time a $C_{ref}$ value has been obtained for the Arctic.

## 1 Introduction

Black carbon (BC) and soot, which originate from incomplete combustion, are particularly potent absorbers of solar radiation and comprise a complex part of the climate system (Bond et al., 2013). Light absorbing particles, including BC and soot, influence the aerosol radiative forcing (ARF) by warming the atmosphere, changing the aerosol scattering albedo and potentially altering cloud droplet





evaporation and lifetime (Koch and Del Genio, 2010). In addition, trace amounts of absorbing particles deposited on snow can perturb snow grain size and thus lower the snow albedo (Hadley and Kirchstetter, 2012; Wiscombe and Warren, 1980a, 1980b); a low albedo favours melting. Polar regions are particularly sensitive to changes in surface albedo which subsequently impacts sea ice, snow cover,

and ultimately surface temperature (Holland and Bitz, 2003; Serreze and Barry, 2011; Serreze et al., 2009). This Polar amplification results in enhanced ice melt and more open water (Johannessen et al., 2004; Serreze et al., 2009). Brown carbon (BrC) absorbs sunlight primarily in the ultraviolet-visible region of the solar spectrum (Andreae and Gelencsér, 2006; Bergstrom and Pilewski, 2007), whereas the BC absorption efficiency is relatively uniform across the UV to near infrared solar spectrum.

Given that BC is a particularly potent perturbing agent, in-situ measurements of BC are important. A widely used technique to measure light absorption by aerosol particles is with filter-based absorption instruments such as the Aethalometer (e.g. Weingartner et al., 2003), the Particle Soot Absorption Photometer (PSAP, Bond et al., 1999; Virkkula et al., 2005), and the Multi Angle Absorption Photometer (MAAP, Petzold and Schönlinner, 2004; Petzold et al., 2005). These instruments report

either equivalent black carbon (eBC) mass concentrations or light absorption coefficients (Petzold et al., 2013).

The high variability of eBC, particularly in Polar, high altitude and coastal regions, makes measurements with Aethalometers challenging. During clean periods, the eBC concentrations can easily be below the detection limit of the instrument. Data treatment methods such as boxcar averaging

can improve the detection limit of the instrument.

An alternative method to reduce noise in Aethalometers has been proposed (Hagler et al. 2011). In this work, a criterion from Hagler et al. (2011) is used, an attenuation change ($\Delta ATN$) threshold needs to be exceeded for post-processing calculations to be invoked. Instead of using this one criterion for boxcar averaging intervals, $\Delta ATN$ is used in the post-processing calculations using the Aethalometer equation.

Here we explore this alternative method from a measurement uncertainty perspective and show that a constant relative uncertainty can be achieved using this one criterion for data post-processing. The result is a time series with a time resolution which is adapted to the measured aerosol concentration. The best performance of this method is achieved when drift in the Aethalometer is at a minimum.

While it is well known that Aethalometer measurements require some form of post-processing (Arnott

et al., 2005; Collaud Coen et al., 2010; Schmid et al., 2006; Virkkula et al., 2007; Weingartner et al., 2003), the purpose of this paper is not to add a correction algorithm to the literature, but to show how to reduce noise in Aethalometer measurements more effectively. This paper uses data from Arctic sites, regions with low signal and high susceptibility to ARF from eBC, to examine noise reduction in the aethelometer signal. Aethalometer instruments have been used to make measurements in the Arctic

since the 1980s (e.g., Bodhaine 1995, Sharma et al., 2006, 2013).

Using the adaptive collection time method of data collection we present an Arctic-specific multiple





scattering enhancement factor ($C_{ref}$) value that can be used in almost all of the existing Aethalometer correction schemes available in literature.

## 2 Measurements and instruments

The data used in this study comprise three years of measurements (2012–2014) at six Arctic stations. The actual eBC climatology of the stations will be presented in a following paper. Below we provide information about station location, operations and environs as well as the instruments deployed at each site. Each site has at least an Aethalometer (variable models) as well as an additional instrument that has been termed the 'reference absorption instrument' for the purposes of this paper.

### 2.1 Measurement sites

#### 2.1.1 Barrow

The Barrow observatory is located on the northernmost coast of Alaska, just 5 km north-east of the town of Barrow, Alaska (population ~4200) and 2 km from Arctic Ocean coast, at an elevation of 11 m asl, and at coordinates 71.323°N and 156.612°W. The site is primarily influenced by regional air

masses originating from the Beaufort Sea, though the station also measures pollution coming from the nearby town. All air masses originating from the direction of the town are marked as contaminated and those data are not used in this analysis.

A 7-wavelength Magee AE31 Aethalometer has been operating at the station since 2010. The reference absorption instrument is the NOAA-built Continuous Light Absorption Photometer (CLAP, Ogren et

al., 2013) that has been collecting aerosol absorption data since 2011. Previous descriptions of the aerosol optical property climatology at BRW can be found in Bodhaine (1983), Bodhaine (1995) and Delene and Ogren (2002).

#### 2.1.2 Alert

Alert is located in Nunavut, Canada, 12 km west of Cape Sheridan, at 82.492° N and 62.508° W, and

at an elevation of 8 m asl. The monitoring station is operated by Environment and Climate Change Canada. Alert is the northernmost site of those analysed here, located just 817km from the North Pole. Given the remote location, the aerosols there are not heavily influenced by human populations. The site is near the coast, which is ice covered in the winter, but turns to open ocean during summer. A 7-wavelength Magee AE31 Aethalometer has been running at Alert from 2008-present. Reference

absorption measurements were made with a 3-wavelength PSAP from 2007-present. More information on black carbon measurements at Alert can be found in Sharma et al. (2002).

#### 2.1.3 Summit

The monitoring station at Summit, Greenland is located at 72.580° N and 38.480° W, and at 3216 m



asl is the highest in elevation of the six sites. Measurements of equivalent black carbon at Summit are supported and operated by Duke University in collaboration with with the NOAA Earth Systems Research Laboratory (ESRL). Although there are many established scientific operations at the Summit site that necessitate activities that produce anthropogenic aerosols, the site is generally very remote

and measures very low aerosol concentrations. Black carbon measurements here have been made with a 1-wavelength (880 nm) Magee AE16 Aethalometer from 2003 to present. The reference absorption instrument at Summit is a multi-wavelength CLAP, running at the site from 2011-present.

### 2.1.4 Zeppelin

The Zeppelin Mountain observatory is located at 475 m asl near the small research village of Ny-

Ålesund on the island of Svalbard at 78.907° N and 11.889° E. The monitoring station is owned by the Norwegian Polar Institute and operated by the Norwegian Institute for Air Research (NILU), and the most recent version of the station building was established in the year 2000. The site is typically located above the inversion layer, and thus measures air masses with minimal contamination. The observatory has long-term measurements of equivalent black carbon with Magee Aethalometers,

namely AE9 from 1998-1999 and AE31 from 2001 to present (Eleftheriadis et al., 2009), and reference absorption measurements with a 1-wavelength PSAP.

### 2.1.5 Pallas

The Pallas measurement station is located in the Finnish Arctic in the Municipality of Muonio. The measurement station is operated by the Finnish Meteorological Institute. The main measurement

building housing the instruments used in this study is located on top of the Sammaltunturi fell. The top of the fell is at an altitude of 565 m asl and above the tree line. The coordinates of the station are 67.973°N 24.116°E. There are no major local sources close to the station and the surrounding terrain is forested, consisting of pine, spruce and birch trees in addition to barren fells.

The 7-wavelength Magee AE31 Aethalometer is connected to the total aerosol inlet which is heated in

order to lower the relative humidity (RH) and causes cloud drops to evaporate. The reference absorption instrument is a MAAP (Thermo Scientific, model 5012). The MAAP is connected to a heated $PM_{2.5}$ inlet to lower the relative humidity. The different size cuts of the instruments could bias the Aethalometer towards higher absorption coefficients than the MAAP. A more thorough description of the site is provided by Hatakka et al. (2003).

### 2.1.6 Tiksi

The Tiksi measurement station is located in northern Siberia in Russia. The station is located 500 metres from the coast of the Laptev Sea at an altitude of 30 m asl at 71.596° N 128.889° E. The site is surrounded by tundra. The station is a cooperation between Russian Federation's Roshydromet, the U.S. National Oceanic and Atmospheric Administration, the U.S. National Science Foundation, and





the Finnish Meteorological Institute. The station is located ≈4 km south of the town of Tiksi, which comprises the sole local source of local air pollution. The data were screened using local wind direction and aerosol size distribution data to omit local pollution from the town (Asmi et al., 2015).

The measurement instruments used in this study consist of a 7-wavelength Aethalometer (Magee model AE31) and a MAAP (model 5012). The instruments are connected to a $PM_{10}$ inlet with self-regulating heating to avoid the build-up of ice on the inlet. By raising the temperature of the sample air to room temperature the sample RH is kept below 30% (Asmi et al., 2015).

## 2.2  Data processing

### 2.2.1 The Aethalometer

The Aethalometer theory of operation relies on the measurement of light transmitted through a fibre-filter as aerosol particles are collected on the filter. The filter is illuminated by a light source from one side with the detectors located on the other side of the filter. Initially, when no aerosol particles have been deposited onto the filter, light is transmitted through the filter with an intensity $I_0$. Then, when aerosol particles deposit onto the filter, the intensity drops to $I$. The Aethalometer calculates, and

reports, filter attenuation ($ATN$) as described in Eq. (1) (e.g. Weingartner et al., 2003).

$$ATN = -100\ln\left(\frac{I}{I_0}\right) \tag{1}$$

The term $I/I_0$ represents the transmission of light through the filter and is referred to as the filter transmittance. The factor of 100 in Eq. (1) is there for numerical convenience and will for this reason also be included throughout this work.    The uncorrected light absorption coefficient ($\sigma_0$) can be written in the form (e.g. Weingartner et al., 2003)

$$\sigma_0 = \frac{A}{Q}\frac{\Delta ATN}{\Delta t} \tag{2}$$

In Eq. (2), $A$ is the filter spot size area, $Q$ is the sample flow rate, and $\Delta t$ is the time between the intensity measurements. The term $\Delta ATN$ is the change in $ATN$ over the time $\Delta t$, which is here called the collection time. When the fibre-filter is loaded with aerosol and the $ATN$ over the filter has dropped too much, the filter spot needs to be changed. In the Aethalometer, filter changes can be set to occur automatically at an $ATN$ value set by the operator. Alternatively, the filter can be set to change after a

given time.

Although the Aethalometer actually measures uncorrected light absorption coefficients ($\sigma_0$), the instrument output is equivalent black carbon (eBC) mass concentration (Petzold et al., 2013). The conversion from $\sigma_0$ to eBC is done using a wavelength-dependent mass attenuation cross-section ($MAC_{AE}$) which is calculated using $14625/\lambda$ $m^2g^{-1}$ (e.g. Arnott et al., 2005), where $\lambda$ is the wavelength

of light in nanometres.

The firmware of the Aethalometer uses an internal collection time $\Delta t$ which is $2 \leq \Delta t \leq 5$ min. This is




the inner data processing cycle of the AE31 Aethalometer. Any longer averaging times set by the operator will commence an outer cycle, which will average the readings obtained during the inner cycle. Therefore, the averaging time ($t_{avg}$) that can be set for the instrument by the operator is restricted to multiples of 5 minutes. In other words, the output of the outer cycle is an average of the inner cycle with a $\Delta t$ of 5 min. This is not always ideal since at very pristine sites a collection time of 5 min is not long enough resulting in noisy data.

Choosing a longer averaging time (the so-called outer cycle) will reduce noise and, therefore, the detection limit of the instrument, at a rate of $t_{avg}^{-0.5}$. Increasing $t_{avg}$, however, results in a reduction of temporal resolution. Moreover, when $t_{avg} > \Delta t$ the instrument output can no longer be reproduced using Eq. (2) since the data that comprise the inner cycle are no longer reported by the instrument. Thus, the greatest versatility of post-processing can be achieved when $t_{avg}$ is equal to $\Delta t$; i.e. $t_{avg} \leq 5$ min for the Aethalometers in this study.

One can circumvent the outer cycle by data post-processing, and achieve a lower detection limit. Included in the standard long-format output of the AE31 and AE16 Aethalometer models are the *ATN* values at the end of the averaging period, along with the aerosol flow rate. Thus, the standard output data can be used to post-process the data using Eq. (2) for an arbitrary value of $\Delta t$; i.e. an arbitrary collection time). The term $\Delta ATN$ is then simply the change in *ATN* from the time $t$ to $t+\Delta t$; i.e. $\Delta ATN = ATN_{t+\Delta t} - ATN_t$.

The benefit of this post-processing approach is that the approach reduces noise better than the boxcar averaging of the firmware. This is discussed and shown further on. This approach for reducing noise in Aethalometer measurements was originally suggested by Hagler et al. (2011). In their work, a $\Delta ATN$ change was used as a criterion for boxcar averaging whereas, here, $\Delta ATN$ is used in the calculations. Previous work using a PSAP has shown that the collection time approach can greatly reduce the noise of filter attenuation measurements (Springston and Sedlacek, 2007) to produce a time series with an adaptive collection time (Hagler et al., 2007). In this work, the method is elaborated on using uncertainty analysis, and specifically for Aethalometers.

## 2.3 Aethalometer multiple scattering correction

As the name suggests, $\sigma_0$ is not the actual light aerosol absorption coefficient of the initially suspended particles. When aerosol particles deposit onto a filter, they will inevitably interact with the filter. The realisation of this has resulted in a variety of different data-processing correction schemes for the Aethalometer (Arnott et al., 2005; Collaud Coen et al., 2010; Schmid et al., 2006; Virkkula et al., 2007; Weingartner et al., 2003). The purpose of these corrections is to derive the actual light absorption coefficient $\sigma_{ap}$ of the suspended particles devoid of filter-induced artefacts.

A fundamental part of these corrections have been the so-called multiple scattering enhancement factor. The multiple scattering enhancement factor



$$C_{\mathrm{ref}} = \frac{\sigma_0}{\sigma_{\mathrm{ap}}} ,$$

(3)

is essentially the amount of light absorption increase caused by the particle-filter interaction after the particles are no longer suspended in air and are deposited on the filter (Weingartner et al., 2003).

Because of multiple scattering, a particle that has deposited onto or into a filter will be subject to non-collimated light. When embedded in a fibre-filter, light absorbing aerosol particles will absorb more

light in the diffuse-light environment in the filter. The diffuse-light environment is due to light being scattered multiple times from fibre to fibre. This effect will diminish as the filter gets loaded with aerosol particles and the optical path of the light through the filter is reduced due to absorbing aerosol particles intersecting scattered light by the fibres.

Furthermore, scattering aerosol will also influence the instrument response; this effect is often named

apparent absorption. The apparent absorption requires a scattering correction, which often is done by subtracting a fixed portion of the scattering coefficients from the absorption coefficients. The intention here is not to add another correction scheme to the literature, but rather to focus on the $C_{\mathrm{ref}}$ value utilized in the existing corrections.

## 2.4  Reference instruments

The Arctic sites in this study were chosen based on the criterion that they all have Aethalometers and an additional reference instrument measuring light absorption coefficients. These additional instruments consist of either a MAAP, PSAP, or CLAP. These instruments will serve as the reference instrument needed to calculate the $C_{\mathrm{ref}}$ for the sites, i.e. they will provide $\sigma_{\mathrm{ap}}$ in Eq. (3).

The MAAP is a filter-based absorption instrument that, in addition to transmittance measurements

through the filter, also measures the back-scattered light at two angles (Petzold and Schönlinner, 2004). This allows for a radiative transfer scheme to be applied since the back-scattered light at multiple angles allows for the distinction between diffusely scattered light and Gaussian scattered light. This information is then used to calculate the diffuse fraction of light scattered back by the filter in order to account for multiple scattering and apparent absorption effects.

The PSAP and CLAP instruments measure transmission, and therefore are based on Eqs (1) and (2). Both instruments use the same type of filter and the optical design of the CLAP is very similar to the PSAP. The CLAP differs from the PSAP in that instead of a single sample spot on a 10 mm filter, it has 8 sample spots on a 47 mm filter. Solenoid valves are used to switch to the next sample spot once the filter transmittance reaches 0.7. Thus, the CLAP can run 8x as long as the PSAP before requiring a

filter change, ideal for remote sites that are not visited daily. The PSAP and CLAP data used in this study were corrected using the Bond et al. (1999) correction along with the Ogren (2010) wavelength adjustment. The Bond correction includes a multiple scattering correction, a filter-loading correction, and an apparent absorption correction. The apparent absorption correction makes use of light





scattering coefficients (e.g., from nephelometers). At all sites in this paper where light scattering coefficients were needed to correct the PSAP and CLAP the light scattering was measured by TSI nephelometers (TSI Inc, model 3563, Anderson and Ogren, 1998).

It should be noted that none of the filter-changes for any of the instruments can be considered to be synchronized with each other, e.g., the PSAP filter is not changed at the same time as an Aethalometer tape advance. Thus, when comparing a reference instrument to an Aethalometer, using the whole time series, any remaining cross sensitivity to the state of the filter on a reference instrument will represent the mean or median bias.

Because the reference instruments operate at different wavelengths than the Aethalometers, Ångström

exponents ($\alpha$) were used to interpolate or extrapolate data to a matching wavelength; $\alpha$ were also used to match nephelometer wavelength to reference absorption wavelengths when using the correction schemes. The Ångström exponent was calculated as follows

$$\alpha = -\frac{\log\left(\sigma_1\right) - \log\left(\sigma_2\right)}{\log\left(\lambda_1\right) - \log\left(\lambda_2\right)} \ , \tag{4}$$

where $\sigma_1$ and $\sigma_2$ represent absorption or scattering coefficients at their respective wavelengths $\lambda_1$ and $\lambda_2$. Using $\alpha$, the absorption coefficient ($\sigma_x$) can be calculated for a desired wavelength $\lambda_x$ using

$$\sigma_x = \sigma_1 \left(\frac{\lambda_1}{\lambda_x}\right)^{\alpha} \tag{5}$$

Using Eqs (4) and (5), the Aethalometer data was interpolated to the wavelengths 467, 525, and 637. The reference absorption instruments that did not already measure at these wavelengths were also interpolated these three wavelengths. The one wavelength Aaethalometer at Summit was interpolated from 880 nm to 637 nm using a $\alpha$ of $-1$ in Eq. (5).

## 3 Aethalometer uncertainty analysis

In order to investigate how the collection time approach can improve the Aethalometer measurements, the measurement uncertainties must be known. By applying the equation for the propagation of uncertainty for uncorrelated variables

$$\delta\sigma_0 = \sqrt{\sum_{i=1}^{n} \left(\frac{\partial \sigma_0}{\partial x_i}\right)^2 \delta x_i^2} \tag{6}$$

to Eq. (2), the relative uncertainty of the measurements can be solved. In Eq. (6), $x_i$ represents the independent variables, $\Delta ATN$, $A$, $Q$, and $\Delta t$ of Eq. (2) and $\delta x_i$ represents their uncertainties.

However, the uncertainty in $\Delta ATN$ has more than one component. Therefore, prior to applying Eq. (6) to Eq. (2) the term $\Delta ATN$ is decomposed into two components. The first component is the true change in $\Delta ATN$ that contains no drift, here denoted as $\Delta ATN_{ND}$. The second component that contributes to the uncertainty in $\Delta ATN$ is drift, here denoted as $\Delta ATN_D$. Furthermore, drift can be expressed as a rate of change over the time $\Delta t$ as $k_D = \Delta ATN_D / \Delta t$. The influence of drift for an arbitrary $\Delta t$ then becomes $k_D \Delta t$.





Thus, $\Delta ATN$ has been decomposed into $\Delta ATN = \Delta ATN_{ND} + k_D \Delta t$. Substituting the total change in $\Delta ATN$ with $\Delta ATN_{ND} + k_D \Delta t$ into Eq. (2) and applying uncertainty propagation (Eq. 6) yields after some rearrangements

$$\frac{\delta \sigma_0}{\sigma_0} = \sqrt{\left(\frac{\delta A}{A}\right)^2 + \left(\frac{\delta Q}{Q}\right)^2 + \left(\frac{\delta \Delta ATN_{ND}}{\Delta ATN_{ND}}\right)^2 + \left(\frac{\delta k_D}{k_D}\right)^2} \ . \tag{7}$$

Note that the term $\delta \Delta t$ has been dropped here since any normal drift in the clock can be neglected.

The determination of both $\delta A$ and $\delta Q$ is to some extent dependent on the instrument operator. The term $\delta A$ can be estimated using a magnifier glass with a scale or digital image analysis to measure the area of the sample spot. Here we will assume that the filter size area can be determined with a 2% uncertainty using digital image analysis.

The value of $\delta Q$ comes from both the accuracy of the calibration and the performance of the flow
controller of the instrument. The uncertainty of the flow meter (Sierra Instruments, Model 824-RFQ-2430) is reported (by the manufacturer) to be 1.5%, which is what will be assumed here. The flow measured by the flow meter is not the exact flow that enters the instrument since there is also a lateral flow through the fibre-filter. The lateral flow will bias the internal flow meter readings towards higher values than the actual flow entering the system. The lateral flow is likely to be a function of the
pressure difference between the sampling line and the room air, which further adds to the uncertainty in the flow rate.

The drift term $(\delta k_D / k_D)^2$ of Eq. (7) is the most demanding to assess as it may vary greatly from station to station for a number of reasons. Drift can be expected to ensue from changes in temperature or relative humidity, changes in lateral flow due to pressure changes in the sampling line, changes in
semi-volatile constituents that have deposited onto the filter, etc. Moreover, lateral flow can influence both the signal and reference detectors, and thus $ATN$, through deposition of aerosol particles that do not originate from the sample air stream. The sources that contribute to drift, and the impact of drift on instrument performance, is best studied under controlled conditions in a laboratory. Therefore, drift will largely be omitted in the uncertainty analysis and discussed on the basis of observations.

By substituting $\delta Q$ in Eq. (7) with the flow rate uncertainty ($f_q$) as a fraction of the total flow $Q$, the term $\delta Q^2$ becomes $(f_q Q)^2$. Equivalently, if the uncertainty of the spot size area ($f_a$) is a fraction of the total area $A$, the term $\delta A^2$ becomes $(f_a A)^2$. Equation (7) then becomes

$$\frac{\delta \sigma_0}{\sigma_0} = \sqrt{f_a^2 + f_q^2 + \left(\frac{\delta \Delta ATN_{ND}}{\Delta ATN_{ND}}\right)^2} \ . \tag{8}$$

Because the drift term has been left out Eq. (8) describes the best case scenario without any drift taken into account. It should be noted that the term $\delta \Delta ATN_{ND}$ describes the random error that originates from





the electronics in the instrument. The relative uncertainty of $\delta\Delta ATN_{ND}/\Delta ATN_{ND}$ can be expressed in terms of measurement-derived values using particle free air as

$$\delta\Delta ATN_{ND,air} = \frac{Q\,\Delta t_{air}}{A}\,\delta\sigma_{0,air} \qquad (9)$$

In Eq. (9), $\delta\sigma_{0,air}$ is the standard deviation of $\sigma_0$ at the time resolution of $\Delta t_{air}$. When determining $\delta\sigma_{0,air}$, $\Delta t_{air}$ should be short so that $\Delta t_{air}$ is the same as the inner cycle for the Aethalometer. Similarly, $\Delta ATN_{ND}$ can be written as a function of $\sigma_0$ and substituted into Eq. (9) which yields

$$\frac{\delta\sigma_0}{\sigma_0} = \sqrt{f_a^2 + f_q^2 + \left(\frac{\delta\sigma_{0,air}\,\Delta t_{air}}{\sigma_0\,\Delta t}\right)^2}\;. \qquad (10)$$

It is often desirable to know the absolute uncertainty ($\delta\sigma_0$) of the measurement in units of the quantity measured. Eq. (10) then becomes

$$\delta\sigma_0 = \sqrt{\sigma_0^2\left(f_a^2 + f_q^2\right) + \left(\frac{\delta\sigma_{0,air}\,\Delta t_{air}}{\Delta t}\right)^2}\;. \qquad (11)$$

Equation (11) implies that the absolute uncertainty of the Aethalometer scales proportionally to $\Delta t^{-1}$ when post-processing using Eq. (2) for a fixed $\Delta ATN$ and no drift; note that $\sigma_0$ inside the square root contains $\Delta t^{-1}$. Solving $\delta\sigma_0$ from Eq. (8) yields the same conclusion.

The $\Delta t^{-1}$ dependency can be verified by measuring particle free air. First a time series of measurements on particle free air is needed. This was obtained by measuring particle free laboratory air with an absolute filter on the inlet of an Aethalometer and logging the extended format of the Aethalometer. Then the drift in $ATN$ was removed by subtracting a running mean of three points producing from the reported $ATN$ values yielding a time series of $ATN$ free of drift ($ATN_{ND}$).

Figure (1) depicts the decomposition of $ATN$ of laboratory measurements when measuring particle free air through an absolute filter. From the figure, it is clear that $ATN$ increased even though no particles could enter the instrument because of the absolute filter connected to the sample inlet of the instrument. This test implies that there can be instrumental drift that only becomes apparent in long time series. $ATN$ and $ATN_{ND}$ shown in Fig. (1a) and (1c) constitute the data used to produce Fig. (2) in addition to the eBC data that was used for the boxcar average $t_{avg}$ in the figure. Fig. (1c) also strengthens the argument that the term $\delta\Delta ATN_{ND}$ is close to the random error from the electronics when using a running mean to derive $\Delta ATN_{ND}$ from $ATN$.

This $ATN_{ND}$ was then used to calculate $\Delta ATN_{ND}$ (and $\Delta ATN$) for a range of $\Delta t$ values (2, 8, 16, 32 … 1024 min) to produce new time series of $\sigma_0$ using Eq. (2). From these time series, the standard deviation of $\sigma_0$ was calculated and plotted as a function of $\Delta t$ as shown in Fig. (2). The time series used comprised 13 days of measurements with a $\Delta t$ of 2 min. Consequently, the values used to calculate $\delta\sigma_0$





in the figure decreased with increasing $\Delta t$.

Figure (2) shows that when the drift is removed the absolute uncertainty $\delta\sigma_0$ follows the predicted $\Delta t^{-1}$ relationship. The logarithmic curve fit for the drift free $\delta\sigma_0$ as a function of $\Delta t$ gives a slope of –1.026. When the drift is not removed, using the running mean method described before, the slope becomes –

0.446. The difference is arguably due to drift. Also shown in the figure is $\delta\sigma_0$ of boxcar averaged $\sigma_0$ converted from the eBC output of the instrument as $\sigma_0 = MAC_{AE} \cdot eBC$. The same time intervals were used for boxcar averaging ($t_{avg}$) as was used for $\Delta t$.

## 4 Measurement results

### 4.1 Measured uncertainties

Aethalometers that are deployed in clean environments can appear at times to just be reporting noise. By simple data post-processing, the signal can be extracted with a greater accuracy, albeit at the expense of temporal resolution (Hagler et al., 2011). This can be done by allowing for a temporal resolution that matches the concentration of species that creates the instrument response, namely the

change in $ATN$, by choosing a constant relative uncertainty (Eq. 8). Equation (8) states that when $f_q$ and $f_a$ are constant, the relative uncertainty depends on the change in filter attenuation ($\Delta ATN_{ND}$). This fact can be used to produce a time series with a constant relative uncertainty.

However, it should be acknowledged that there is an additional uncertainty due to instrument drift but in principle a constant uncertainty could also be achieved using Eq. (7). That would require a thorough

investigation into the sources for the drift and how they vary between stations, not feasible in this study given the remote locations of the stations. However, based on the laboratory measurements the drift can be significant on a time scale from hours to days. When the aim is to determine the drift at a station, the absolute filter should be attached to the sampling line to capture the pressure changes in the sampling line relative to ambient pressure, changes in relative humidity, for different aerosol types

and filter loadings, and light absorption coefficients of the air that constitute the lateral flow through the instrument.

The measurement uncertainty for the six Aethalometers at the respective stations were determined by measuring particle free air. For all stations except Alert, particle free air was sampled for at least 24 hours with an absolute filter connected to the instrument inlet. These measurements are shown in Fig.

(3) at a wavelength of 590 nm (Summit 880 nm). For Alert, the particle free air was sampled for a few hours per week comprising 4 days of data in total.

Figure (3) depicts the $ATN$ drift in the different Aethalometers during the particle free air measurements. The figure shows that all the tested Aethalometers experienced drift during the particle free air measurements. Also evident from the figure is that the drift of the different Aethalometers (and





different sites) can be very different. Based on the figure, it is not enough to conduct measurements on particle free air for a few hours in order to assess the instrument performance at the site. Filtered air measurements should instead be performed over a period of 24 hours, or more. These measurements should be conducted on a pristine filter to minimize the influence of semi-volatile constituents that

have deposited onto the filter (Cappa et al., 2008; Lack et al., 2008).

For reference, a linear drift of $\Delta ATN_\mathrm{D} = 1.0$ in 24 hours is shown in Fig. (3) which corresponds to a $\sigma_0$ value of 0.07 Mm$^{-1}$ when using $Q$=5 lpm, $A$=0.5 cm$^2$, and $\Delta t$=24 h. The consequence of a linear drift of 1.0 in 24 hours would also set the lowest value possible to achieve. As can be seen from the inserts of Fig. (1), $ATN_\mathrm{D}$ need not be increasing all the time, and thus lower values of $\sigma_0$ are possible to

achieve during periods with little drift. It should be mentioned that this drift will also affect the eBC concentrations reported by the instrument. For the five instruments evaluated here, the drift uncertainty is seen in Fig. (2) to be roughly 0.01–0.1 Mm$^{-1}$.

The standard deviations of the $\Delta ATN_\mathrm{ND}$ measurements made with an absolute filter in-line are shown in Table 1. Because the only wavelength dependent variable in Eq. (8) is $\delta\Delta ATN_\mathrm{ND}$, the change in the

$\Delta ATN$ measurements with respect to wavelength will also be the sole source of the difference in the relative uncertainty between different wavelengths. The values that describe the relative uncertainty in in terms of $\Delta t$ and $\sigma_0$ (Eq. 10) are presented in Table 2.

Figure (4) shows the relative uncertainty ($\delta\sigma_0/\sigma_0$) of Eq. (8) as a function of $\Delta ATN_\mathrm{ND}$ based on measurements conducted with an absolute filter upstream of the instrument. For clarity, the figure was

produced using a mean of all wavelengths to represent the typical relative uncertainty of the instrument. The mean values were calculated from Table 1. Figure (4) shows how the relative uncertainty decreases when $\Delta ATN$ increases. The upper y-axis scale of $\sigma_0$ in the figure was calculated for reference using $Q$=5 lpm, $A$=0.5 cm$^2$, and $\Delta t$=60 min.

Implicit from both Fig. (4) and Eq. (8), is that the relative uncertainty of the instrument changes with

the aerosol concentration when using a fixed $\Delta t$; for a fixed $\Delta t$, $\Delta ATN$ will change according to the concentration. The equation for the relative uncertainty (Eq. 8) can be used as a criterion to achieve a more constant level of uncertainty which was not captured when the method was introduced by Hagler et al. (2011). This can either be determined from Fig. (4) directly, or calculated from Eq. (8) after the term $\delta\Delta ATN_\mathrm{ND}$ has been determined.

One way to characterize the performance of an Aethalometer is to calculate the $\Delta ATN$ value at which the flow ($f_\mathrm{q}$) and spot size ($f_\mathrm{a}$) uncertainties together are equally important as the $\Delta ATN$ uncertainties. This is shown in Table 3. The crossover was calculated by solving $\Delta ATN$ from the terms under the square root of Eq. (8), namely $\Delta ATN_\mathrm{ND}=(\delta\Delta ATN_\mathrm{ND}^2/(f_\mathrm{q}^2+f_\mathrm{q}^2)^2)^{1/2}$. The uncertainty in the flow rate, relative to the uncertainty in the $ATN$ measurements, diminishes exponentially when $\Delta ATN$ decreases





(Fig. 4). Here, a criterion of $\Delta ATN \geq 2$ was used in the post-processing of the data to also allow for a lower detection limit in the boxcar averaged reference data that is discussed in the next section.

For the sake of simplicity, this criterion was only applied to the middle wavelength of the Aethalometer (590 nm). If the criterion were to be applied to all wavelengths, one would end up with seven time series for each instrument, with different timestamps. That could make further data analysis unnecessarily convoluted. It is worth pointing out that if the data set being analysed is going to be averaged, then the $ATN$ values included with the averaged data set should not be averaged—an averaged $ATN$ would make $\Delta t$ less well-defined. Instead, either the first or the last $ATN$ value during the averaging period should be incorporated into the averaged data set.

The time series of the one hour boxcar averaged Aethalometer data is show in Fig. (5). The time series of the adaptive collection time is shown in Fig. (6). When using the adaptive collection time, it is clear that when the absorption coefficient is low, the time resolution is low. At higher absorption coefficients, the time resolution is better. This is desirable since it means that at high concentrations of light absorbing aerosol particles there is no loss in temporal resolution whereas at low concentrations, this adaptive method is capable of reaching lower detection limits quicker than boxcar averaging when drift is minimal.

Comparing Figs (5) and (6), it is clear that the adaptive collection time approach is to be favoured when $\sigma_{ap} < 0.1$ Mm$^{-1}$ because of instrument noise. In Fig. (5), it is shown that at low $\sigma_{ap}$ the one hour averages of the data set are clearly more scattered than when using the adaptive collection time method when $\sigma_{ap}$ is low (Fig. 6). Since the y-scale of Fig. (5) is logarithmic, negative values are not shown, although they are still present in the one hour averaged time series. By definition, the adaptive collection time approach will not produce negative $\sigma_0$ values since $\Delta ATN$ is always positive.

In fact, for the measurements studied here, when the $\sigma_0$ is above 2.1–6.7 Mm$^{-1}$, there is no loss in the temporal resolution in the one hour averaged data of Fig. (5). The range in $\sigma_0$ is due to the fact that the different Aethalometers at these 6 Arctic sites are operated at different flow rates. Figure (7) shows histograms of $\Delta t$ for the different stations using the adaptive collection time approach.

Figure (6) shows values that are lower than the example drift in $\Delta ATN = 1.0$ in 24 hours (Fig. 3), which implies that there are periods where the drift can be substantially lower. In Figs (5) and (6), the $\sigma_{ap}$ values come from $\sigma_0$ values that have been corrected for multiple scattering correction using a $C_{ref}$ value of 3.10 as discussed in the next section. Thus, the drift uncertainty seen in Fig (3) becomes 0.003–0.03 Mm$^{-1}$ after multiple scattering correction is applied.

## 4.2 Multiple scattering enhancement in the Arctic

Multiple scattering correction factors have been reported for a range of different sites around the





world, but none have focused primarily on the Arctic. The multiple scattering correction values $C_{ref}$ reported in the literature are summarized in Table 4.

Some of the variations in the reported $C_{ref}$ values of Table 4 can be attributed to the different ways in which they were calculated. Some $C_{ref}$ values were calculated with a filter loading correction applied and some without. Moreover, some of the $C_{ref}$ values were calculated with both a scattering and a filter loading correction applied. For the sake of inter-comparability, a $C_{ref}$ value calculated from unaltered raw data, without any of the available correction algorithms, would be preferable. Here, $C_{ref}$ values are calculated for the 6 Arctic sites using Eq. (3), where $\sigma_{ap}$ is the absorption coefficient from the reference absorption instrument and $\sigma_0$ is the uncorrected absorption coefficient obtained from the Aethalometer using Eq (2). Equation (3) is the same as Eq. (11) in Weingartner et al. (2003), with the exception that here $\sigma_0$ is not the value at an *ATN* value of 10, but for the full range of *ATN* values.

There are several possible issues with the derivation of $C_{ref}$ values presented here. First, in this study, the reference absorption measurements also rely on measurements using filter-based absorption measurements—it remains unclear to which extent this will affect the absolute values of $C_{ref}$ because no absorption standard measurements were available at the sites. However, since the filter changes of the different instruments are not synchronized, and because the data sets cover three years at each site, it can be assumed that there is very little coincidence with respect to filter loading effects. Thus, the $C_{ref}$ values presented here should represent typical values for the different sites. Second, the flow rates of the different instruments differ which can affect the $C_{ref}$ values due to different penetration depths (Lack et al., 2009; Nakayama et al., 2010). Third, it has to be acknowledged that there can be a bias in the absolute $C_{ref}$ values because of imperfect corrections of filter artefacts in the reference instruments (Backman et al., 2014; Müller et al., 2011). However, this bias should not substantially alter the *ATN* dependency because filter changes were not performed in sync. For a pristine filter, the sole artefact should be $C_{ref}$ (Collaud Coen et al., 2010). As the filter gets loaded with aerosol particles, loading effects comes in to play. These loading effects change between the filter spots depending on the optical properties of the aerosol that is being deposited on that particular spot (Virkkula et al., 2015), and even during sampling on the same spot (Drinovec et al., 2015).

Such detailed analysis of filter loading effects is not feasible with this data set since it would require data with a high temporal resolution and preferably concurrent non-filter-based light absorption measurements. In general, the goodness of evaluation for all filter-based light absorption measurements should be continuous light absorption coefficients over filter spot changes so that a filter spot change would go unnoticed; this should hold true for all aerosol types and loadings. This means that there would not be an *ATN* dependency when compared to non-filter-based light absorption measurements.

It has been shown that published Aethalometer correction algorithms, which aim to compensate for filter loading and multiple scattering effects, do not necessarily remove the *ATN* dependence when





applied on data from different stations (Fig. 4 in Collaud Coen et al., 2010). Again, the aim is not to add another correction algorithm to literature. Instead, the $C_{ref}$ values presented here should be interpreted as a means to make Aethalometers in the Arctic more inter-comparable by introducing a $C_{ref}$ value for the Arctic using the reference instruments at hand.

Figure (8) shows the calculated $C_{ref}$ values as a function of $ATN$ for the six Arctic sites. Since the data depicted in Fig. (8) were produced using a concentration-adapted collection time, the statistics in the figure were calculated using a collection time weighted percentile (Hyndman and Fan, 1996). Without this weighting, the statistics would have effectively been concentration weighted. Figure (8) is equivalent to Fig. (4) of Collaud Coen et al. (2010) for the $C_{ref}$ values labelled "AE manufacturer" in

their figure.

In general, Tiksi and Pallas show the highest $C_{ref}$ values, whereas Summit shows the lowest. Summit stands out as an outlier in Fig. (8); it is the station at the highest elevation and uses 1-wavelength Aethalometer (880 nm). The Summit Aethalometer data were interpolated to a wavelength of 637 nm using a $\lambda^{-1}$ wavelength dependence. A summary of the different $C_{ref}$ values calculated for the stations is

presented in Table 5.

In addition to the different $C_{ref}$ values observed over the $ATN$ range in Fig. (8), there are other differences among stations. Some of the $C_{ref}$ values decrease as a function of $ATN$. In the $ATN$ range of 0–10, the median $C_{ref}$ values for Alert and Tiksi are larger than at the other stations, but at higher $ATN$, the Alert and Tiksi $C_{ref}$ values decrease. This is expected behaviour and is due to the filter loading

effect causing a decrease in Aethalometer sensitivity. However, a decrease in $C_{ref}$ with $ATN$ is barely noticeable for the Barrow and Zeppelin data sets, although the variation in $C_{ref}$ at Zeppelin makes the trend—or lack thereof—less clear.

Again, Summit shows a different behaviour altogether. As the filter $ATN$ increases, so do the $C_{ref}$ values. This is contrary to the expected behaviour of the filter loading effect in which loading

generally decreases the sensitivity of a filter-based absorption measurement technique (Arnott et al., 2005; Virkkula et al., 2007). The filter loading effect is most pronounced for an aerosol with a low single-scattering albedo ($\omega_0$) (Sheridan et al., 2005). $\omega_0$ is defined as the ratio of light-scattering ($\sigma_{sp}$) to light extinction ($\sigma_{sp}+\sigma_{ap}$); $\omega_0 = \sigma_{sp}/(\sigma_{sp}+\sigma_{ap})$. The fact that Summit does not follow this trend suggests that the aerosol optical properties of Summit are different in relation to the other stations. The

different behaviour, however, does not seem to be related to $\omega_0$ as the $\omega_0$ of Summit does not stand out. The difference could be due to different asymmetry parameter or size distribution of the aerosol at Summit. The Pallas Aethalometer was operated with a filter change interval of 8 hours for most of the time, and therefore there are very few data points with an $ATN$ above 10. Hence, it is questionable if there are enough data points to be able to draw conclusions about a trend in the $C_{ref}$ and $ATN$

relationship at Pallas.

In the upper part of Table 5, it can be seen that there is not much variation in the $C_{ref}$ values at different





wavelengths. In the table, the Aethalometer data were interpolated using Ångström exponents to match the wavelength of the reference instruments using Eqs (4) and (5). It cannot be ruled out that the low variability in $C_{ref}$ with wavelength is a consequence of the reference instruments used. However, long-term measurements of aerosol absorption from non-filter-based measurement instruments co-located

with Aethalometer measurements are not available in the Arctic. The lower part of Table 5 provides statistics of $C_{ref}$ for each site. Because there does not seem to be a great wavelength dependence on the $C_{ref}$ value at Alert and Barrow, the $C_{ref}$ value for all wavelengths was calculated using all available wavelengths. The overall value of $C_{ref}$ for sites across the Arctic was determined to be 3.10. The value of 3.10 was calculated using average-time weighted median as discussed earlier, and the weighted 25[th]

and 75[th] percentiles for the all wavelength $C_{ref}$ values are 2.56 and 3.78, respectively. Because Summit appears to be significantly different from the other Arctic stations, the Summit $C_{ref}$ was omitted from the grand median $C_{ref}$ calculation.

Figure 9 depicts the relationship between the reference absorption instrument ($\sigma_{ap}$) and the uncorrected absorption coefficient ($\sigma_0$) which yields $C_{ref}$; see Eq. (3). The figure is provided as an overview

comprising all wavelengths and all stations; except for Summit because of the reasoning mentioned before. In the figure, in addition to the weighted median value of 3.10, the slope of the bivariate fit is also shown. The fit was performed using bivariate regression with the averaging time as weights (Cantrell 2008). The slope of the bivariate regression becomes 3.24. Both the regression method and the weighted median method yield values that are within 5% of each other. The systematic root-mean-

square error (RMSE) was calculated from the predicted $\sigma_0$ using the results from the regression, which becomes 0.81 Mm$^{-1}$. The standard deviation (STD) was calculated using the standard error (SE) of the bivariate fit and the number of data points ($n$) as STD = SE·$\sqrt{n}$ = 1.30 Mm$^{-1}$. The mass absorption cross-section (MAC) describes the relationship between eBC mass concentrations and $\sigma_{ap}$. Similarly, MAC$_{AE}$ describes the relationship between $\sigma_0$ and eBC as given by the manufacturer; both MAC and

MAC$_{AE}$ have units of m$^2$g$^{-1}$. A simple evaluation can be performed to investigate whether the $C_{ref}$ value is reasonable, assuming that the difference between MAC$_{AE}$ and MAC is $C_{ref}$ (Arnott et al., 2005). If the Aethalometer measured at a wavelength of 550 nm, then MAC$_{AE}$ would be 26.59 m$^2$g$^{-1}$. Compensating MAC$_{AE}$ with $C_{ref}$=3.10 would yield a MAC of 8.6 m$^2$g$^{-1}$; i.e. a MAC which is 3.10 times lower than MAC$_{AE}$. This MAC is within the range suggested by Bond and Bergstrom (2006), namely

7.5±1.2 m$^2$g$^{-1}$ at 550 nm, which implies that the $C_{ref}$ value determined here is reasonable. This simple evaluation, however, does not take into account any apparent absorption or coating effects.

## Conclusions

In clean environments, such as in the Arctic during summer months, measurements of aerosol light absorption coefficients can be below the detection limit of the instrument. Symptomatically, it is not

uncommon to encounter measurements reporting negative equivalent black carbon concentrations or light absorption coefficients. These values are without physical meaning and originate from instrument





noise and uncertainties.

Here a post-processing method for Aethalometer data based on collection time is elaborated on. This post-processing approach allows for an arbitrary collection time $\Delta t$, which lowers the electronic noise of the Aethalometer proportionally to $\Delta t^{-1}$. In comparison, boxcar averaging lowers the noise proportional to $t_{avg}^{-0.5}$. The greatest benefit of this approach can be achieved when drift in the Aethalometer *ATN* measurements is minimized.

The noise characteristics of Aethalometers are best estimated using measurements of particle free air. Based on these measurements, it is recommended that particle free air measurements should be conducted for at least 24 hours, or more. Furthermore, the absolute filter used for particle free air measurements should be connected between the instrument and the sampling line from which the sample is drawn during normal operation. From these data, the electronic noise and drift can be evaluated.

The uncertainty analysis showed that the collection time approach can be used with a simple criterion that keeps the signal-to-noise ratio constant, namely that the post processing calculations are invoked once the filter attenuation of the instrument has changed by more than $\Delta ATN$. This criterion will cause the collection time to vary according to the concentration of absorbing aerosol particles. The collection time approach was applied to Aethalometer data from 6 Arctic monitoring sites using $\Delta ATN \geq 2$.

In addition, using co-located 'reference' absorption measurements at each site, an Arctic specific multiple scattering enhancement factor ($C_{ref}$) was calculated using the collection time approach as described above. For all wavelengths, and all low-altitude Arctic stations (i.e., all stations except Summit), the median $C_{ref}$ value was calculated to be 3.10. The 25[th] to 75[th] percentile range of $C_{ref}$ was 2.56–3.78. The $C_{ref}$ value for Summit was calculated to be 1.57. The reason for the low $C_{ref}$ value at Summit remains unresolved.

### Acknowledgements

This work was supported by the Academy of Finland project Greenhouse gas, aerosol and albedo variations in the changing Arctic (project number 269095) and the Academy of Finland Center of Excellence program (project number 272041). John Backman wishes to acknowledge Maj and Tor Nessling foundation grant 2014044 and 201600449 for financial support. We acknowledge Russel Schnell for providing Aethalometer data from Summit. Authors would like to acknowledge Alert operators for lab and instrument maintenance and CFS Alert for the logistics and operations of Alert base camp. More generally, the authors would like to acknowledge the personnel an researchers at the respective measurement stations that have contributed to the data. We acknowledge the Aerosol working group of the International Arctic System for Observing the Atmosphere (IASOA) for coordinating the data and expert contributions to this work.

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



## Tables

**Table 1:** Standard deviations of $\delta\Delta ATN_{ND}$ for the different Aethalometers and their measurement wavelengths. The filtered air noise measurements consist of at least 24 hours of data; except for Alert where data was comprised a few hours of measurements totalling 4 days. The standard deviation was calculated from subsequent reported $ATN$ values as such and can therefore be used to reproduce Fig. (4). The $t_{avg}$ column shows the instrument setting for the outer cycle of the instrument during the time of the noise measurements.

| | N | $t_{avg}$ | 370 | 470 | 520 | 590 | 660 | 880 | 950 |
|---|---|---|---|---|---|---|---|---|---|
| | | min | | | | nm | | | |
| **Alert** | 1267 | 5 | 0.011 | 0.010 | 0.011 | 0.010 | 0.011 | 0.008 | 0.008 |
| **Summit** | 235 | 5 | | | | | | 0.016 | |
| **Barrow** | 745 | 5 | 0.016 | 0.016 | 0.015 | 0.015 | 0.015 | 0.015 | 0.015 |
| **Tiksi** | 316 | 5 | 0.007 | 0.008 | 0.006 | 0.007 | 0.007 | 0.074 | 0.004 |
| **Pallas** | 290 | 5 | 0.003 | 0.004 | 0.003 | 0.004 | 0.003 | 0.003 | 0.004 |
| **Zeppelin** | 48 | 30 | 0.028 | 0.012 | 0.010 | 0.031 | 0.016 | 0.012 | 0.014 |

**Table 2:** Standard deviation of $\sigma_0$ ($\delta\sigma_0$) in $Mm^{-1}$ when measuring particle free air. These values can be used in Eq. (10) for an arbitrary value of $\sigma_0$ and $\Delta t$.

| | $\Delta t$ | 370 | 470 | 520 | 590 | 660 | 880 | 950 |
|---|---|---|---|---|---|---|---|---|
| | min | | | | nm | | | |
| **Alert** | 5 | 0.284 | 0.251 | 0.286 | 0.253 | 0.282 | 0.215 | 0.213 |
| **Summit** | 5 | | | | | | 0.283 | |
| **Barrow** | 5 | 0.332 | 0.325 | 0.316 | 0.312 | 0.322 | 0.313 | 0.318 |
| **Tiksi** | 5 | 0.137 | 0.155 | 0.117 | 0.129 | 0.151 | 1.519 | 0.086 |
| **Pallas** | 5 | 0.136 | 0.171 | 0.144 | 0.149 | 0.111 | 0.114 | 0.156 |
| **Zeppelin** | 30 | 0.058 | 0.026 | 0.021 | 0.065 | 0.032 | 0.024 | 0.029 |

**Table 3:** Crossover $\Delta ATN$ above which the flow rate uncertainty ($f_q$=1.5%) and spot size uncertainty ($f_a$=2.0%) together become more important than $\delta\Delta ATN_{ND}$.

| | 370 nm | 470 nm | 520 nm | 590 nm | 660 nm | 880 nm | 950 nm |
|---|---|---|---|---|---|---|---|
| Alert | 0.87 | 0.77 | 0.87 | 0.77 | 0.86 | 0.65 | 0.65 |
| Summit | | | | | | 1.27 | |
| Barrow | 1.27 | 1.25 | 1.21 | 1.19 | 1.24 | 1.20 | 1.22 |
| Tiksi | 0.54 | 0.61 | 0.46 | 0.51 | 0.59 | 5.96 | 0.34 |
| Pallas | 0.26 | 0.33 | 0.28 | 0.29 | 0.21 | 0.22 | 0.30 |
| Zeppelin | 2.23 | 1.00 | 0.82 | 2.49 | 1.25 | 0.94 | 1.11 |





**Table 4:** Summary table of different $C_{ref}$ values reported in literature for different types of locations and therefore aerosol types.

| Site | Site or aerosol type | $C_{ref}$ | Citation |
|---|---|---|---|
| Las Vegas, USA | Urban | 3.69 | Arnott et al. (2005) |
| Laboratory | Diesel soot | 2.09 – 2.22 | Weingartner et al. (2003) |
| Amazon, Brazil | Biomass burning | 5.23 | Schmid et al. (2006) |
| Jungfraujoch, Switzerland | Free troposphere | 2.8 – 7.77 | Collaud-Coen et al. (2010) |
| Cabauw, Netherlands | Polluted continental | 4.09 – 4.57 | Collaud-Coen et al. (2010) |
| Mace Head, Ireland | Coastal | 3.05 – 3.83 | Collaud-Coen et al. (2010) |
| Hohenpeissenberg, Germany | Rural continental | 2.78 – 3.16 | Collaud-Coen et al. (2010) |

**Table 5:** Multiple scattering enhancement ($C_{ref}$) for the different stations. The values were calculated using averaging-time weighted percentiles because of the adaptive average time used to derive them. The top portion of the table reports the $C_{ref}$ values for all available wavelengths of the reference absorption instruments. Aethalometer wavelengths were interpolated to these reference absorption wavelengths using absorption Ångström exponents. The Summit AE-16 data were extrapolated to a wavelength of 637 nm using $\alpha=-1$. The bottom portion of the table reports the statistics of $C_{ref}$ using all available wavelengths. The last row in the table shows the number of data points used for the statistics.

| | Alert | Summit | Barrow | Tiksi | Pallas | Zeppelin | Overall[†] |
|---|---|---|---|---|---|---|---|
| | | | $C_{ref}$ for individual wavelengths | | | | |
| **467 nm** | 3.29 | - | 2.94 | - | - | - | - |
| **525 nm** | 3.27 | - | 2.88 | - | - | 2.98 | - |
| **637 nm** | 3.27 | 1.57 | 2.90 | 3.67 | 3.66 | - | - |
| | | | Percentile values of $C_{ref}$ (all wavelengths) | | | | |
| **25th** | 2.71 | 0.77 | 2.41 | 3.11 | 2.93 | 2.21 | 2.56 |
| **50th** | 3.28 | 1.57 | 2.91 | 3.67 | 3.66 | 2.98 | 3.10 |
| **75th** | 4.01 | 2.40 | 3.36 | 4.34 | 5.01 | 5.54 | 3.78 |
| **N** | 3244 | 1024 | 3391 | 2290 | 3160 | 2563 | 14648 |

[†]The overall statistics comprise all stations except the high altitude station of Summit.





**Figures**

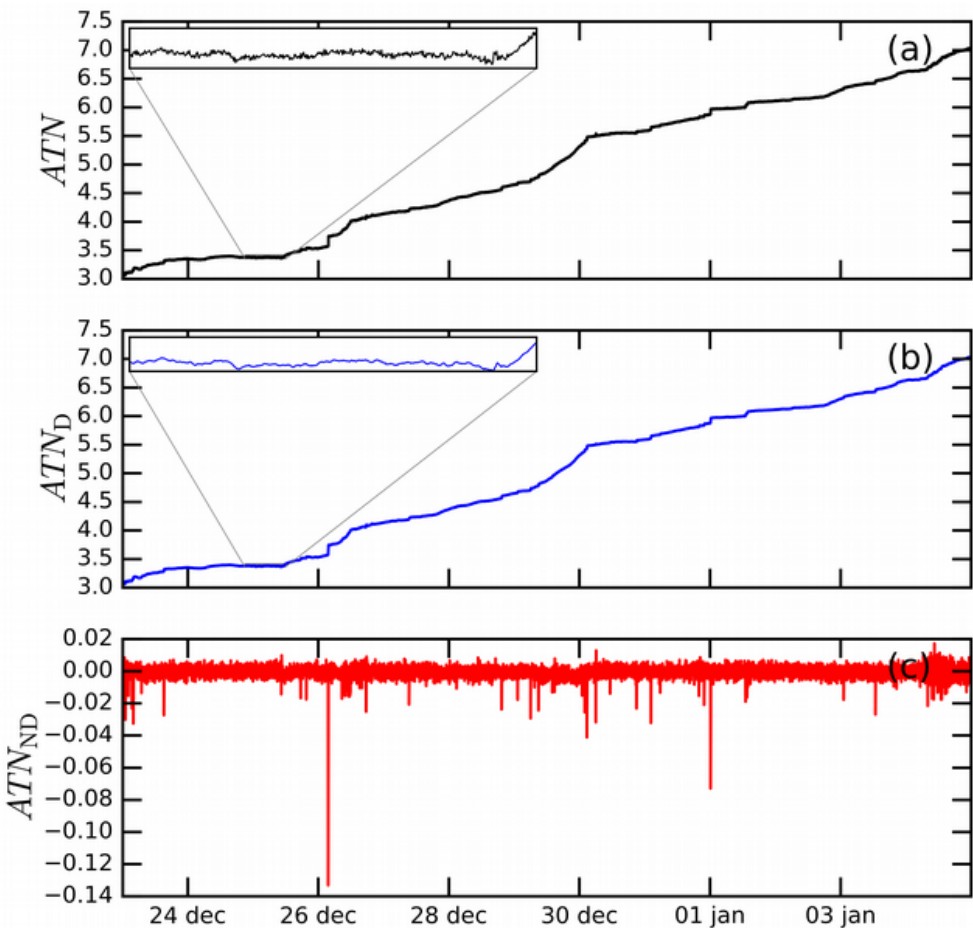

**Figure 1:** Decomposition of *ATN* from measurements of particle free air at a wavelength of 520 nm. Panel **(a)** shows the *ATN* values as reported by the instrument. Panel **(b)** shows the 3 point running mean which and represents the drift in *ATN* ($ATN_D$). Panel **(c)** shows the $ATN$-$ATN_D$ which is free of drift ($ATN_{ND}$).





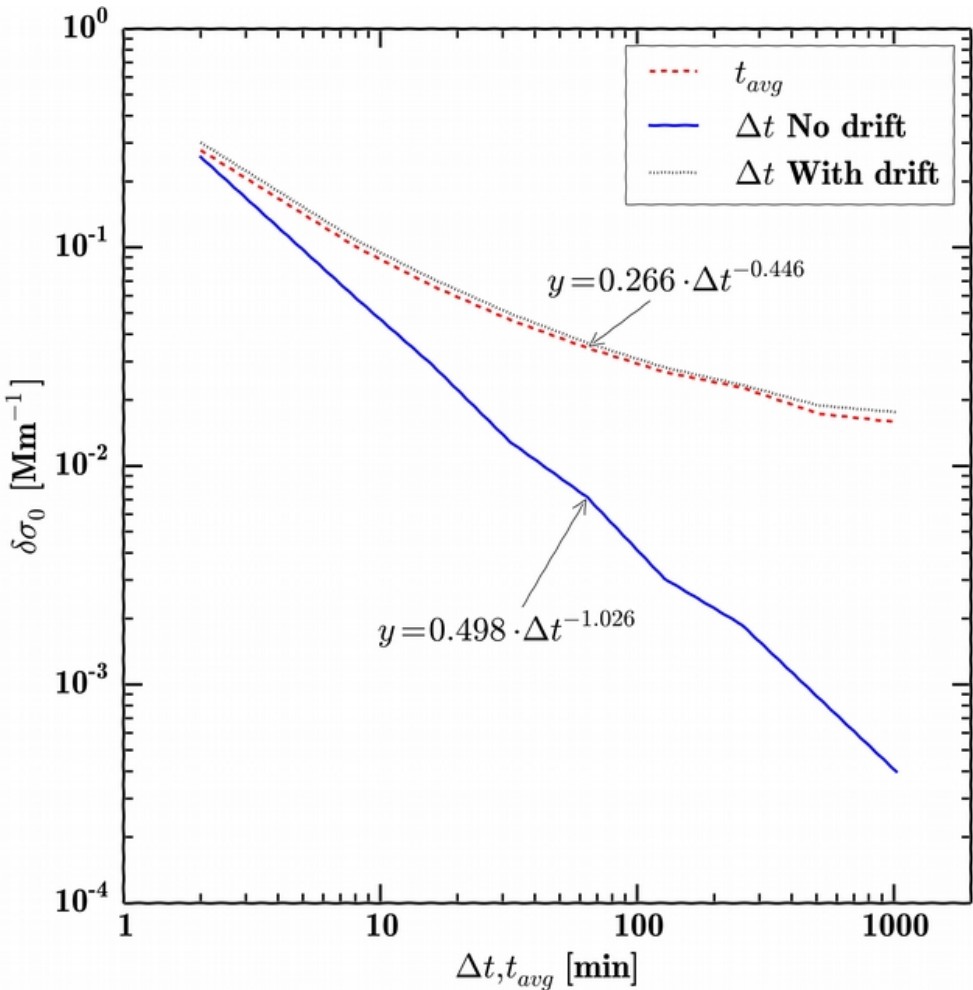

**Figure 2:** Standard deviation of uncorrected light absorption coefficients ($\delta\sigma_0$) when measuring particle free air as a function of collection time ($\Delta t$) with drift and without drift in the data. The $t_{avg}$ curve is calculated from eBC data as reported by the instrument and converted to $\sigma_0$ using a $MAC_{AE}$ of 28.13 m²g⁻¹. The wavelength used to produce the figure is 520 nm.



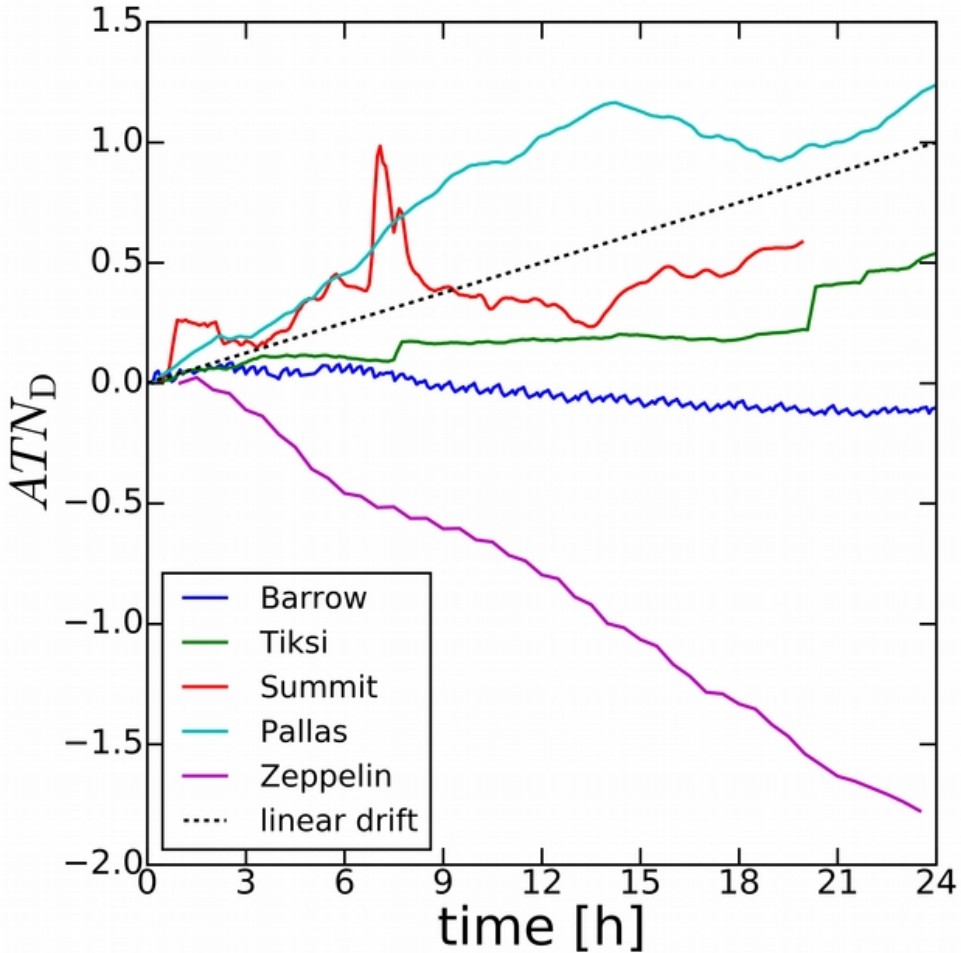

**Figure 3:** The drift in *ATN* (*ATN*$_D$) during measurements of particle free air at five arctic station. The linear drift shown in the figure corresponds to a $\sigma_0$ value of 0.07 Mm$^{-1}$ when Q= 5 lpm, A=0.5 cm$^2$, and the drift in *ATN* is 1 in 24 hours. In the figure, *ATN*$_D$ has been forced to begin at 0% for easier comparison.





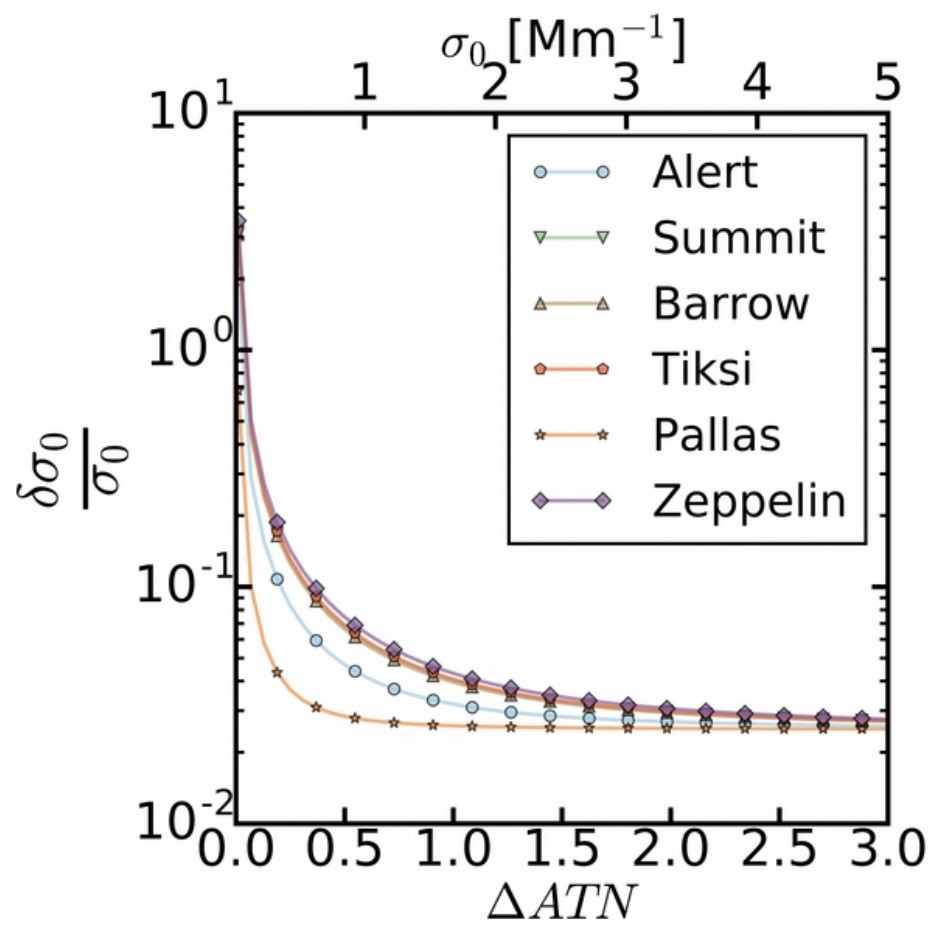

**Figure 4**: Relative uncertainty of uncorrected absorption coefficients as a function of change in filter attenuation ($\Delta ATN$); see Eq. (7). The upper x-scale was calculated using $A$=0.5 cm2, $Q$=5 lpm and $\Delta t$=60 min for reference.

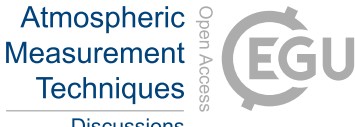



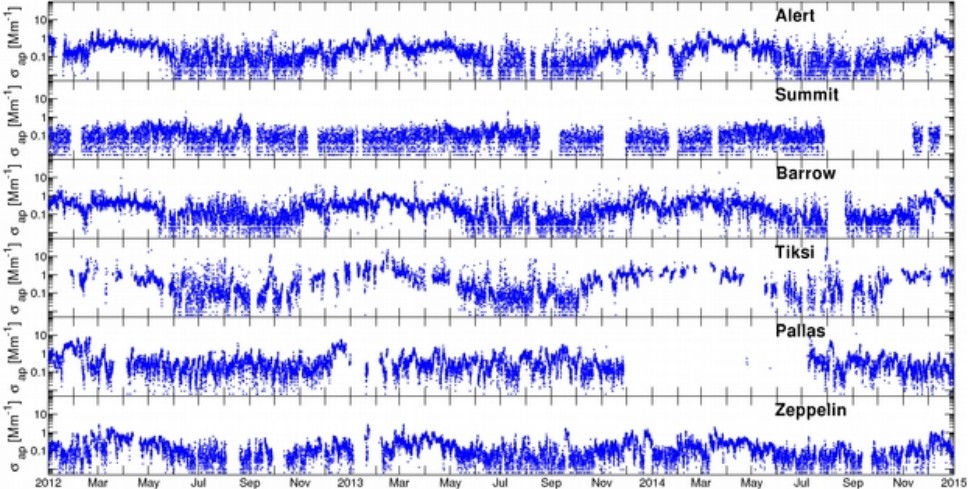

**Figure 5:** Time series of one hour averaged absorption data for $\lambda = 520$ nm. In the figure, the absorption coefficients have been corrected for using the multiple scattering artifact $C_{ref} = 3.10$.

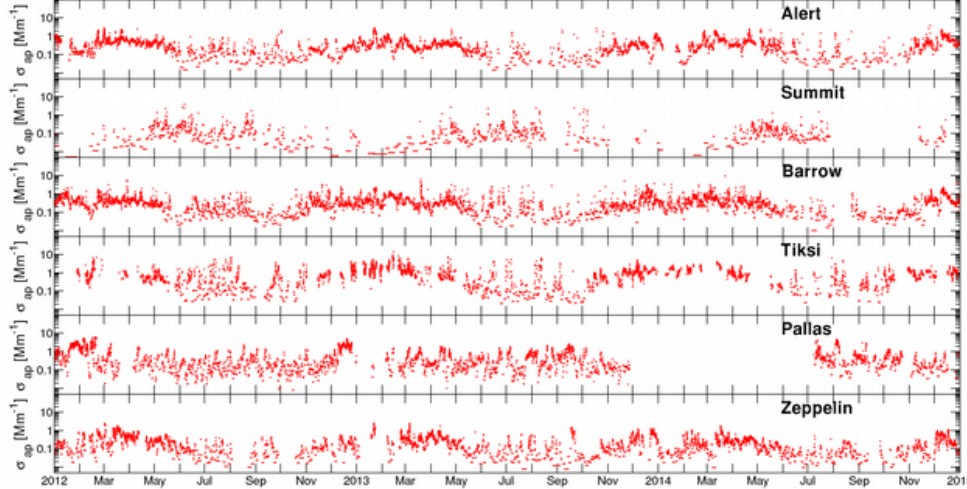

**Figure 6:** Time series of absorption coefficients using the adaptive collection time approach at a wavelength of 520 nm. The absorption coefficients have been corrected for using the multiple scattering artifact $C_{ref} = 3.10$.





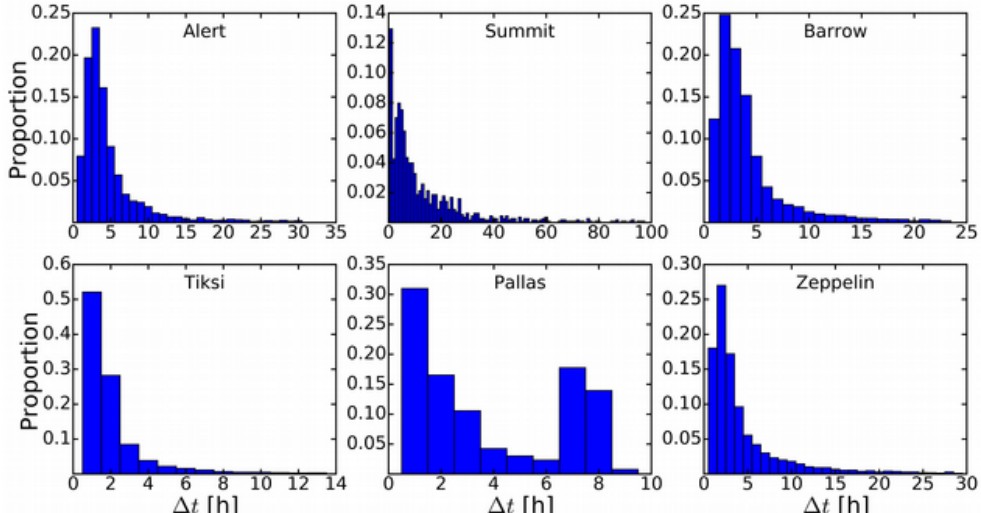

**Figure 7:** Normalized histogram of the collection time $\Delta t$ for the different stations for a $\Delta ATN$ threshold of 2.

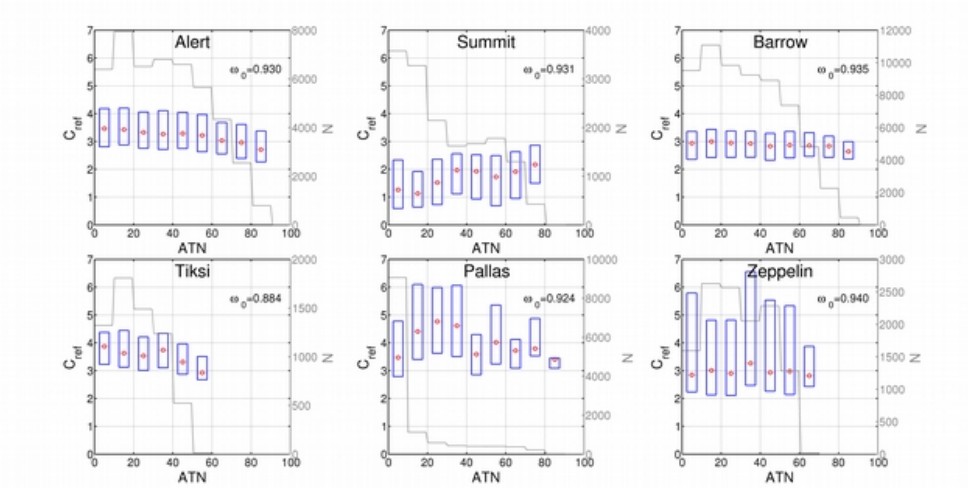

**Figure 8:** Multiple scattering enhancement ($C_{ref}$) as a function of filter attenuation ($ATN$) calculated using Eq. (3). The dashed line and the right hand y-axis show the number of data points that each $ATN$ range comprise. The blue boxes represent the 25[th] to 75[th] percentile range whereas the red diamonds represent the median values. $C_{ref}$ values in the figure are for all available wavelengths. The figure also shows the median $\omega_0$ of the aerosol using the absorption coefficients from Fig. (6).





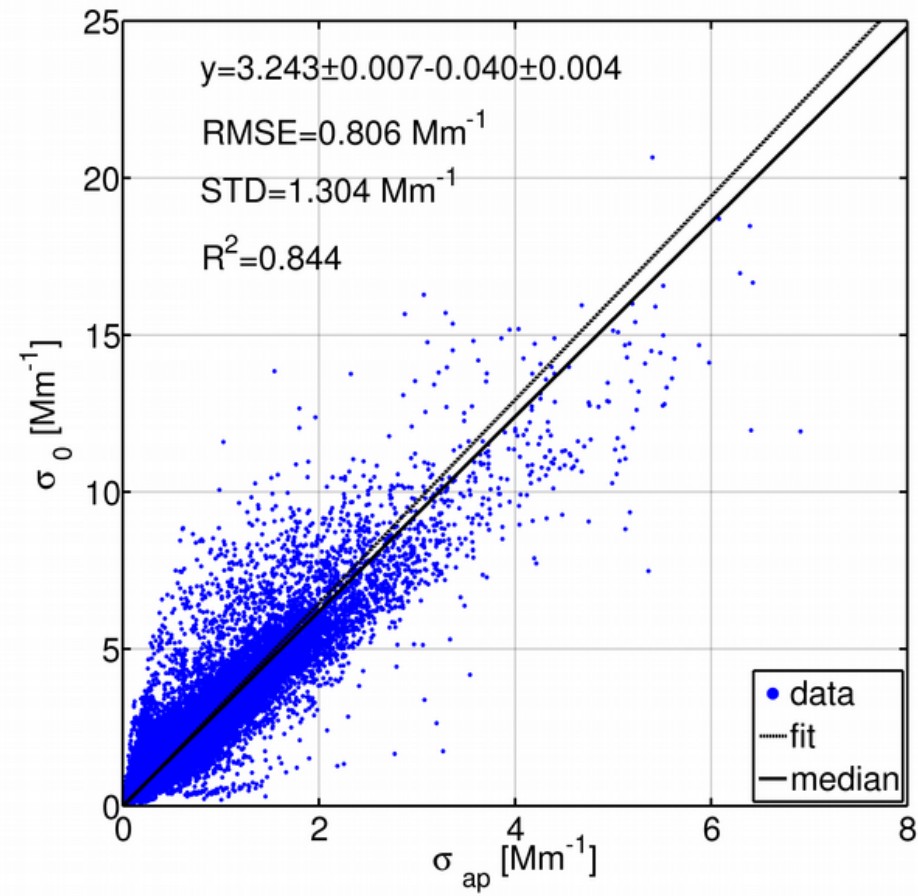

**Figure 9:** The figure shows the uncorrected absorption coefficients ($\sigma_0$) as a function of reference absorption coefficients ($\sigma_{ap}$) including all available wavelengths and all stations; except for Summit. To this data, using bivariate regression, a first order polynomial was fitted using averaging times as weights; shown as a dotted line. The solid line marks the weighted median $C_{ref}$ value of 3.10. In the figure, RMSE stands for systematic root-mean-square error and STD for standard deviation. The standard deviation was calculated using the standard error (SE) of the fit and number of data points ($n$) as STD=SE·$\sqrt{n}$. $R^2$ is the correlation coefficient.