# Peer review of "On Aethalometer measurement uncertainties and multiple scattering enhancement in the Arctic"

_Atmospheric Measurement Techniques, 2016_

## Referee Comment (RC1) · Anonymous Referee #2 · 9 Jan 2017

Review on ' On Aethalometer measurement uncertainties and multiple scattering enhancement in the Arctic' by J. Backman et al.

General comments

The articles deals with two issues of filter based absorption photometers: First, the reduction of noise by data post-processing, and second, an empirical determination of an enhancement factor due to multiple scattering. These issues have physically different origins and are not related. Anyway, it is justified to present both one article. The noise reduction is essential for deriving multiples enhancement factors in environments with low ambient absorption coefficients as it is the case for Artic aerosols.

[Figure]

The article presents new insights in characterising filter based absorption photometers and gives good advices to other scientists for instrument setup and data processing in environments with low aerosol concentrations. The existence of scattering enhancement factors is a known fact in the scientific community. It is the first time that enhancement factors were derived for the Artic.

The paper is well written and clearly structured. The content meets the requirements to be published in AMT. The reviewer suggests the article for publication with minor corrections.

Specific comments

Page 9, line 5: The authors estimate an uncertainty of determining the filter spot area to be 2%. Is that the actual uncertainty derived from the spot to spot variability or is it the precision of the measurement of a single spot area. Were time series corrected for systematic deviations of the spot size (cf. Eq. 2) as described in Bond et al. (1999) for the PSAP.

Note: Page 7 line 26 ff.: The reviewer thinks that it needs to be better explained why data from CLAP are corrected using the "Bond" corrosion. One possibility could be as follows. PSAP and CLAP use the same type of filter and similar design. Nakayama et al. (2010) showed that similar instruments, COSMO and PSAP, with the same type of filter have a similar size-dependent scaling factor. A similar size dependent scaling factor is a hint that the overall correction could be similar. Are there other publications showing that CLAP and PSAP are similar in the literature?

Page 7, line 16 ff: Why is it justified to use PSAP, MAAP, and CLAP as reference instruments? On page 14 line 13 the authors wrote "...the reference absorption measurements also rely on measurements using filter-based absorption measurements - it remains unclear to which extend this will affect the absolute value of C_ref ...". The reviewer agrees that a full analysis of the error of C_ref under this circumstances is not possible. Anyway, the question still is what are the advantages of PSAP, MAAP, and

CLAP compared to Aethalometers and what are the reasons for using this instruments as reference. Otherwise, C_ref would merely be a harmonisation factor for comparing results from instruments of different type.

Chapter 4.2 The reviewer thinks that estimated uncertainties and detection limits for the reference instruments should be presented along with that values for the Aethalometers.

The reviewer is wondering how the noise reduction algorithm affects the final results, the value and uncertainty of C_ref.

Technical corrections

The article is well written. Figures and tables are of good quality. There is no need for technical corrections.

---

## Referee Comment (RC2) · Anonymous Referee #1 · 29 Jan 2017

The work presented in the manuscript is very important as it introduces a new technique for post-processing filter photometer data, focusing on the Aethalometers in the Arctic. The novel approach synthesizes previous work on the reduction of instrumental noise and the associated detection limit. The manuscript is detailed and presents the post-processing approach in a way that will render the methodology useful for users of different filter photometers: Aethalometers, PSAPs, CLAPs. . . The approach is novel in that it allows averaging the data to obtain constant relative uncertainty by changing the averaging window. It also shows the very low uncertainties which can be obtained using this methodology. The presented work is an important contribution to the interpretation of Arctic and global measurements of aerosol absorption. In addition, the

authors report the relative scaling factor between the raw Aethalometer measurements of the attenuation coefficient and the processed PSAP/CLAP/MAAP measurements, reporting the absorption coefficient. They report this as the scattering enhancement factor.

There are terminological and methodological issues that need to be accounted for prior to publication in AMT.

Once the major and minor issues are addressed, the manuscript would make a perfect addition to the compendium of filter photometer related literature in AMT.

The measurement in filter photometers such as Aethalometer, PSAP and CLAP is one of transmission of light and the determination of the change of attenuation (ATN). Then the attenuation coefficient is calculated and the eBC concentration is derived from this coefficient using the mass attenuation cross-section. While the authors use the proper term "mass attenuation cross-section" in the text, they use the term "uncorrected light absorption coefficient (sigma_0)" (starting on p. 5, line 20). This is inaccurate – the quantity is the "attenuation coefficient", this quantity is then post-processed for loading effects and divided by the multiple scattering coefficient C to obtain the "absorption coefficient". This procedure is based on assumptions which need to be tested to the greatest extent possible.

The authors use the PSAP, CPAL and MAAP as "reference" instruments. The claim of "reference" is not substantiated. The paper very clearly delineates the way to obtain the factor C, but this can be interpreted just as the relative normalization factor to harmonize the determination of the absorption coefficient from different filter photometers. And here lies the crux of the problem: all instruments which are being compared are filter photometers and the principle of operation for most of them is nearly identical. The claim of C being interpreted as the "multiple scattering enhancement factor" needs to be further substantiated. Since no non-filter method was available, the methodology needs to be proven at least internally consistent.

[Figure]

The determination of the absorption coefficient necessitates the determination of the multiple scattering parameter C (Weingartner et al., 2003). The parameter C is to a degree arbitrarily separated from the loading effects, which influence the determination of the absorption coefficient as well. If C is to be the parameter describing the multiple scattering effects in the filter matrix, it should not depend on ATN. This can be considered to be the "proof" of the separation of the multiple scattering from the loading effects (the Weingartner et al. parameter R). The authors show that C does depend on ATN (Fig. 8, p. 29). The authors correctly point out that the existing post-processing algorithms do not necessarily ensure the lack of dependence of C on ATN (Collaud Coen et al., 2010), however for background sites, the loading effects are most probably non-existent (Virkkula et al., 2015; Drinovec et al., 2016). The post-processing algorithm needs to be site specific, as the loading effects are a function of the entire loading of the sample spot and the physical and chemical properties of the entire sample deposit. This could be the reason of the difference between Summit and other Arctic sites.

The reason for the C dependence on ATN can be due to the non-compensation of the Aethalometer data (even though this is questionable for global background sites; Virkkula et al., 2015; Drinovec et al., 2016) or the loading effects in the so-called "reference" instruments, which are known to feature loading effects or saturation (Bond et al., 1999; Virkkula et al., 2005; Hyvärinen et al., 2013). The authors correctly identify this weakness of the presented work in the beginning of section 4.2 when they mention that the C they report is essentially a slope between attenuation and absorption coefficients determined with different filter photometers.

The authors need to present the criteria for the "goodness of evaluation" of loading effects in all filter photometers. They have already used the way to go about this (when discussing C): the aerosol absorption should not depend on ATN for measurements in all filter photometers. This analysis should be added to section 4.2 and discussed: the plot of sigma_abs=sigma_abs(ATN) for all sites. The relationship between the C and the scattering coefficient should be reported – do scattering particles in the filter

increase the attenuation coefficient?

In addition to this, the Aethalometers are compared to different instruments: two different versions of PSAP, CLAP and MAAP. The authors need to substantiate that the comparison of the Aethalometers to these different instruments is relevant. No comparison between the PSAPs, CLAP and MAAP is reported. The authors should at least sum up the results of laboratory inter-comparisons if no comparisons for ambient Arctic measurements is available.

Specific comments

Page 2, line 22: "below the detection limit". As the authors later point out, the detection limit is a function of the time between two consecutive measurements and the averaging time. One can lower the detection limit by integrating the sample for a longer time. The sentence needs to be modified or the time resolution (5 min?) needs to be specifically mentioned.

P. 3-5: it would be a good idea to report the inlet cuts (PM2.5. . .), flows (or face velocities), operational wavelengths of the filter photometers for all sites. The conditions for triggering the change of tape should also be reported (8 hours in Pallas, for example, elsewhere an ATN limit).

P. 5, l. 14: "Initially, when no aerosol particles have been deposited onto the filter, light is transmitted through the filter with an intensity I_0." This is not true, the Aethalometers measure I_0 (intensity of light transmitted through the reference part of the filter without any sample) at the same time as the intensity of light I transmitted through the sample. Please change.

P. 5, l. 20 and repeated later: "uncorrected light absorption coefficient ($\sigma 0$)". This is the "attenuation coefficient", please see above. Please change accordingly.

P. 5, l. 31: "14625/$\lambda$ m2g–1". Please use unitless expressions, for example "16.6 m2g–1 at 880 nm, scaling inversely with the wavelength" or something similar.

P. 7. Section 2.4: the comparison of the so-called "reference" instruments needs to be elaborated (please see above). Has there been an intercomparison of the instruments in question? In ambient conditions or in the laboratory? How are published results of other intercomparisons relevant for the results reported in the manuscript?

P. 8., l. 9: is the 2% relative uncertainty for spot area specific to Aethalometers or to all filter photometers. Can the diameter be measured to 0.1 mm? How do relative and absolute uncertainties of the "reference" filter photometers influence the results?

P. 8, l. 19: "The one wavelength Aaethalometer at Summit was interpolated from 880 nm to 637 nm using a $\alpha$ of $-1$ in Eq. (5)." Please correct the typo "Aaethalometer" -> "Aethalometer". What is the real value of the absorption Angstrom exponent $\alpha$? Could the lower C determined at Summit be (partly) due to the systematic bias due to the extrapolation from 880 nm to 635 nm using an $\alpha$ value that is too small? Are there any multi-wavelength measurements of absorption at Summit (sat least a short time series)?

P. 9, l. 22: "Moreover, lateral flow can influence both the signal and reference detectors, and thus ATN, through deposition of aerosol particles that do not originate from the sample air stream." This is highly unlikely, as the particles will get filtered out at the edges of the filter material, not above the light detectors in the filter photometers. Please substantiate the claim or remove the sentence.

P. 10-11, Fig. 1: The sources of the drift in the laboratory experiment with the absolute filter are intriguing. The authors should at least offer some hypothesis on them and comment whether they are relevant for ambient measurements. How does the additional pressure drop due to the filter influence the measurements? Does it in fact introduce additional drift? Are there jumps or transients when tape is moved and measurements restarted? Do semi-volatile organic compounds adsorb on the filter and cause a signal, appearing as drift? Is the air conditioning in the laboratory important? Does drift have a wavelength dependence (indicating sample deposition rather than a

fixed electronic drift)?

P. 11-12, Fig. 3: Was the absolute filter attached to the Aethalometer or to the sampling line leading to the Aethalometer? Were other instruments attached to the same sample line? Were there any pressure drops, jumps, transients in the sample line, or events in the measurement room, resulting in movement of the filter in the Aethalometers? Any movement or drift in the filter position influences the measurement of ATN. The authors need to comment on these possible sources of drift and the observed transients.

P. 13, l. 32-33: "Thus, the drift uncertainty seen in Fig (3) becomes 0.003–0.03 Mm–1 after multiple scattering correction is applied." The effect of the reduction of noise on the uncertainty of C and the effect of thus derived C on the uncertainty of the absorption coefficient should be presented more clearly, starting perhaps here.

P. 14, Table 4: The determination of the C within ACTRIS "grey literature" reports would be a valuable reference here.

P. 14, l. 22-25: "Third, it has to be acknowledged that there can be a bias in the absolute Cref values because of imperfect corrections of filter artefacts in the reference instruments (Backman et al., 2014; Müller et al., 2011). However, this bias should not substantially alter the ATN dependency because filter changes were not performed in sync." This is an oversimplification. At constant concentration and equally spaced movements of tape (but not synchronized, and dependent on the flow of the individual instruments) an back-side-of-an-envelope calculation shows that the effect can be up to 20%. This is unlikely for global background sites in the Arctic, but this needs to be shown at least in the Supplement of the manuscript.

P. 15, Figure 9: The regression equation on the figure is missing sigma_ap as the independent variable, making it confusing for the reader.

References Arnott, W. P., Hamasha, K., Moosmuller, H., Sheridan, P. J., and Ogren, J. A.: Towards aerosol light-absorption measurements with a 7-wavelength Aethalometer:

[Figure]

evaluation with a photoacoustic instrument and 3-wavelength nephelometer, Aerosol Sci. Technol., 39, 17-29, doi:10.1080/027868290901972, 2005.

Bond, T., Anderson, T. L. and Campbell, D.: Calibration and intercomparison of filter-based measurements of visible light absorption by aerosols, Aerosol Sci. Technol., 37–41, 1999.

Collaud Coen, M., Weingartner, E., Apituley, A., Ceburnis, D., Fierz-Schmidhauser, R., Flentje, H., Henzing, J. S., Jennings, S. G., Moerman, M., Petzold, A., Schmid, O., and Baltensperger, U.: Minimizing light absorption measurement artifacts of the Aethalometer: evaluation of five correction algorithms, Atmos. Meas. Tech., 3(2), 457–474, doi:10.5194/amt-3-457-2010, 2010.

Drinovec, L., Gregorič, A., Zotter, P., Wolf, R., Bruns, E. A., Prévôt, A. S. ~H., Petit, J.-E., Favez, O., Sciare, J., Arnold, I. J., Chakrabarty, R. K., Moosmüller, H., Filep, A., and Močnik, G.: The filter loading effect by ambient aerosols in filter absorption photometers depends on the mixing state of the sampled particles, Atmos. Meas. Tech. Discuss., doi:10.5194/amt-2016-285, in review, 2016.

Hansen, A. D. A., Rosen, H., and Novakov, T.: Real-time measurement of the aerosol absoprtion-coefficient of aerosol particles, Appl. Opt., 21, 3060–3062, doi: 10.1364/AO.21.003060, 1982. Hansen, A. D. A., Rosen, H., and Novakov, T.: The aethalometer – an instrument for the real-time measurement of optical absorption by aerosol particles, Sci. Total Environ., 36, 191–196, 1984.

Hyvärinen, A.-P., Vakkari, V., Laakso, L., Hooda, R. K., Sharma, V. P., Panwar, T. S., Beukes, J. P., van Zyl, P. G., Josipovic, M., Garland, R. M., Andreae, M. O., Pöschl, U., and Petzold, A.: Correction for a measurement artifact of the Multi-Angle Absorption Photometer (MAAP) at high black carbon mass concentration levels, Atmos. Meas. Tech., 6, 81-90, doi:10.5194/amt-6-81-2013, 2013.

Virkkula, A., Ahlquist, N. C., Covert, D. S., Arnott, W. P., Sheridan, P. J., Quinn, P.

[Figure]

K. and Coffman, D. J.: Modification, Calibration and a Field Test of an Instrument for Measuring Light Absorption by Particles, Aerosol Sci. Technol., 39(1), 68–83, doi:10.1080/027868290901963, 2005.

---

## Author Comment (AC1) · 31 Mar 2017

**Reply to comment by Anonymous Reviewer 2**

The authors wish to thank the referee for the constructive comments and criticism.

**Specific comments**

**Comment:** Page 9, line 5: The authors estimate an uncertainty of determining the filter spot area to be 2%. Is that the actual uncertainty derived from the spot to spot variability or is it the precision of the measurement of a single spot area. Were time series corrected for systematic deviations of the spot size (cf. Eq. 2) as described in Bond et al. (1999) for the PSAP.

**Reply:** The unit-to-unit variability is greater than what the spot size measurement uncertainty of 2%. The 2% is the uncertainty by which the spot size can be measured. We see no need to change the sentence as this should be clear as it stands. Both the PSAP and CLAP instruments were spot size corrected as described by Bond et al. (1999). The Aethalometers were not spot-size corrected because we do not have filter spots to measure on from all the instruments. We do not have filter spots from the MAAPs either.

**Comment:** Note: Page 7 line 26 ff.: The reviewer thinks that it needs to be better explained why data from CLAP are corrected using the "Bond" corrosion. One possibility could be as follows. PSAP and CLAP use the same type of filter and similar design. Nakayama et al. (2010) showed that similar instruments, COSMO and PSAP, with the same type of filter have a similar size-dependent scaling factor. A similar size dependent scaling factor is a hint that the overall correction could be similar. Are there other publications showing that CLAP and PSAP are similar in the literature?

**Reply:** The explanation is that both instruments use the same type of filters; namely the Pallflex type E70-2075W filter. This is the same argumentation why the COSMOS also uses the Bond et al. (1999) correction (Miyazaki et al., 2008). A sentence was added that both the PSAP and CLAP use the same type of filter. Moreover, an additional sentence was added stating that "It has been shown before that the same type of filter and a similar optical design yield very similar results (Miyazaki et al. 2008; Nakayama et al. 2010)". There are only grey literature reports on the intercomparison between PSAP and CLAP instruments such as the one cited and ACTRIS reports (http://www.wmo-gaw-wcc-aerosol-physics.org/files/actris-intercomparison-workshop-integrating-nephelometer-and-absorption-photometer-02-03-2013.pdf). The most relevant report is already cited.

**Comment:** Page 7, line 16 ff: Why is it justified to use PSAP, MAAP, and CLAP as reference instruments? On page 14 line 13 the authors wrote "...the reference absorption measurements also rely on measurements using filter-based absorption measurements it remains unclear to which extend this will affect the absolute value of Cref ...".
The reviewer agrees that a full analysis of the error of Cref under this circumstances is
not possible. Anyway, the question still is what are the advantages of PSAP, MAAP, and
CLAP compared to Aethalometers and what are the reasons for using this instruments
as reference. Otherwise, Cref would merely be a harmonisation factor for comparing
results from instruments of different type.

**Reply:** The choice of PSAP, MAAP and CLAP as 'reference' instruments is prag-
matic; no other instruments were available. The shortcomings of the 'reference' instru-
ments are even more clearly stated in the revised manuscript. Moreover, the revised
manuscripts make it even more clear that $C_{\text{ref}}$ (changed to $C_{\text{f}}$) is a harmonisation factor
for the comparison of Aethalometers with the co-located filter-based absorption pho-
tometers. We also clearly state that $C_{\text{f}}$ should not be literally interpreted as a multiple
scattering correction factor but merely a harmonising correction factor. This criticism is
thoroughly discussed in the replies to the general comments by the other anonymous
referee.

**Comment:** Chapter 4.2 The reviewer thinks that estimated uncertainties and detection
limits for the reference instruments should be presented along with that values for the
Aethalometers.

**Reply:** The absolute uncertainties of PSAP and MAAP are now stated in Section 4.2
and compared to values in Table 3 which comprise the absolute uncertainties of the dif-
ferent Aethalometers. Since we did not perform zero air measurements on the MAAP,
PSAP and CLAP instruments the discussion is restricted to previously published val-
ues.

**Comment:** The reviewer is wondering how the noise reduction algorithm affects the
final results, the value and uncertainty of Cref.

**Reply:** Three additional paragraphs were added to the beginning of section 4.2 dis-
cussing this topic in more detail. This was also picked up by the other referee.

**Additional references**

Miyazaki, Y., Kondo, Y., Sahu, L. K., Imaru, J., Fukushima, N. and Kano, M.: Performance of a newly designed continuous soot monitoring system (COSMOS), J. Environ. Monit., 10(10), 1195, doi:10.1039/b806957c, 2008.

---

## Author Comment (AC2) · 31 Mar 2017

**Reply to comment by Anonymous Referee 1**

The authors wish to thank the referee for the constructive comments and criticism.

**General comments**

**Comment:** The measurement in filter photometers such as Aethalometer, PSAP and CLAP is one of transmission of light and the determination of the change of attenuation (ATN). Then the attenuation coefficient is calculated and the eBC concentration is derived from this coefficient using the mass attenuation cross-section. While the

authors use the proper term "mass attenuation cross-section" in the text, they use the term "uncorrected light absorption coefficient (sigma_0)" (starting on p. 5, line 20). This is inaccurate – the quantity is the "attenuation coefficient", this quantity is then post-processed for loading effects and divided by the multiple scattering coefficient C to obtain the "absorption coefficient". This procedure is based on assumptions which need to be tested to the greatest extent possible.

**Reply:** The term "uncorrected light absorption coefficient ($\sigma_0$)" is also used for the attenuation coefficient (e.g. Virkkula et al. 2005). The referee is right that attenuation coefficient is more descriptive for $\sigma_0$ in the contexts of this work. The term uncorrected absorption coefficient was replaced with attenuation coefficient throughout the manuscript as suggested. To obtain light absorption coefficients from attenuation coefficients has been the topic of a multitude of articles and cannot be considered resolved. It is true that this procedure requires assumptions. In order to scrutinise these assumptions, true reference instruments are needed. Because of the lack of true reference instruments, this work has to rely on existing correction algorithms.

**Comment:** The authors use the PSAP, CPAL and MAAP as "reference" instruments. The claim of "reference" is not substantiated. The paper very clearly delineates the way to obtain the factor C, but this can be interpreted just as the relative normalization factor to harmonize the determination of the absorption coefficient from different filter photometers. And here lies the crux of the problem: all instruments which are being compared are filter photometers and the principle of operation for most of them is nearly identical. The claim of C being interpreted as the "multiple scattering enhancement factor" needs to be further substantiated. Since no non-filter method was available, the methodology needs to be proven at least internally consistent.

**Reply:** The authors do not claim that PSAP, CLAP or MAAP would be ideal reference instruments. This is clearly stated in section 4.2. In addition, this is now also stated in the earlier section of 2.4. The choice of PSAP, MAAP and CLAP as 'reference' instruments is purely pragmatic since no other instruments exist at the stations.

The claim of 'reference' cannot be substantiated beyond the instruments shortcomings of which a substantial proportion of section 4.2 is dedicated to. To make it more clear to the reader, "reference instrument" was changed to "co-located absorption photometer" as not to give the false impression of them being independent reference instruments.

It is true that the term 'multiple scattering enhancement factor' is problematic when other filter-based measurements are the only available instruments for comparison. The referee is correct that the reported $C_{\text{ref}}$ can be considered to be a relative normalisation factor to harmonise the determination of the absorption coefficient from different Aethalometers. This is the intention. This harmonisation has the additional benefit that attenuation or light absorption coefficients from different sites can be compared to measurements using the same type of instrument. In the light of the criticism on the use of the term "multiple scattering enhancement factor" in the general criticism of the manuscript by the reviewer, the term was rephrased to just "correction factor $(C_{\text{f}})$" although "multiple scattering enhancement factor" has been used in literature before for the comparison of Aethalometers measurements to absorption coefficients as measured by MAAPs (Collaud Coen et al., 2010). The title was also changed to "On Aethalometer measurement uncertainties and an instrument correction factor for the Arctic"

**Comment:** The determination of the absorption coefficient necessitates the determination of the multiple scattering parameter C (Weingartner et al., 2003). The parameter C is to a degree arbitrarily separated from the loading effects, which influence the determination of the absorption coefficient as well. If C is to be the parameter describing the multiple scattering effects in the filter matrix, it should not depend on ATN. This can be considered to be the "proof" of the separation of the multiple scattering from the loading effects (the Weingartner et al. parameter R). The authors show that C does depend on ATN (Fig. 8, p. 29). The authors correctly point out that the existing post-processing algorithms do not necessarily ensure the lack of dependence of C on ATN (Collaud Coen et al., 2010), however for background sites, the loading effects are most probably non- existent (Virkkula et al., 2015; Drinovec et al., 2016). The postprocessing algorithm needs to be site specific, as the loading effects are a function of the entire loading of the sample spot and the physical and chemical properties of the entire sample deposit. This could be the reason of the difference between Summit and other Arctic sites.

**Reply:** The authors agree that $C_{ref}$ parameter is arbitrarily separated from the loading effects. A more precise term for $C_{ref}$ could be 'initial multiple scattering enhancement factor' because multiple scattering cannot be distinguished from other effects when the filter gets loaded multiple scattering still occurs in a loaded filter; although reduced with increased loading. Thus it could also be argued that they cannot be separated.

The authors agree that, in general, site-specific post-processing for filter and transmission-based absorption photometers would be beneficial when light absorption coefficients are desired from attenuation coefficients. The need for, or the tuning of, existing correction algorithms during data post-processing should aim to remove any $ATN$ dependence originating from loading artefacts and this should be the motivation for correcting attenuation coefficients during post-processing. As the reviewer points out, an $ATN$ dependence on any spot, at any time, depends on the properties of the pre-deposited aerosol on that spot. No filter spots are alike. Therefore, loading corrections would ideally be done on a spot to spot basis which is not possible in the Arctic due to low concentrations. A generally tuned site-specific correction would not guarantee that there is no $ATN$ dependency; it would just on average be $ATN$ independent.

The difference between Summit and the other sites can be due to different physical and chemical properties of the aerosol. This is discussed in section 4.2 to the extent that is possible given the instruments at hand. To extend the analysis to physical and chemical properties (beyond optical means) would be well outside the scope of this manuscript. Another reason for the mismatch of Summit to the other sites is the fact that the Summit Aethalometer is a non-standard AE-16 that has been converted from a broadband light source to a LED light source. Summit is also the only AE16 model aethalometer. This information was added to the manuscript. A paragraph was also added on scattering Ångström exponents which is indicative of aerosol particle

size distribution to discuss the difference between Summit and the other sites. In addition, based on the referee's specific comments, the interpolation to the 'reference' instruments wavelength was done using the average wavelength dependence of the reference instrument at the site. This did not explain the difference. Furthermore, as addressed in the specific comments, an $ATN$ dependence of $\sigma_{\mathrm{ap}}$ from a co-located instrument is not expected to fundamentally change the observed $ATN$ dependence.

**Comment:** The reason for the C dependence on ATN can be due to the non-compensation of the Aethalometer data (even though this is questionable for global background sites; Virkkula et al., 2015; Drinovec et al., 2016) or the loading effects in the so-called "reference" instruments, which are known to feature loading effects or saturation (Bond et al., 1999; Virkkula et al., 2005; Hyvärinen et al., 2013). The authors correctly identify this weakness of the presented work in the beginning of section 4.2 when they mention that the C they report is essentially a slope between attenuation and absorption coefficients determined with different filter photometers.
The authors need to present the criteria for the "goodness of evaluation" of loading effects in all filter photometers. They have already used the way to go about this (when discussing C): the aerosol absorption should not depend on ATN for measurements in all filter photometers. This analysis should be added to section 4.2 and discussed: the plot of sigma_abs=sigma_abs(ATN) for all sites. The relationship between the C and the scattering coefficient should be reported – do scattering particles in the filter increase the attenuation coefficient?

**Reply:** Again, we do not claim that the 'reference' instruments would be free from loading effects. The filter changes in the PSAP and CLAP 'reference' instruments were changed at a transmittance of 0.7 according to Bond et al. (1999) in order to keep loading effects minimal and MAAP filters were changed at a transmittance of 50% in Pallas and 20% in Tiksi. What we do claim, however, is that MAAPs do not suffer from saturation given the low concentrations in the Arctic. Saturation in the MAAP only occurs at very high (>0.04 $\mu$g min$^{-1}$) equivalent black carbon (eBC) mass accumulation

rates (Hyvärinen et al., 2013). Moreover, the saturation is a firmware effect and is not related to the measurement technique itself.

The goodness of evaluation is presented in the manuscript as was requested by the referee already in the quick report. To prove or disprove an $ATN$ dependency of all filter-based photometers to the level of certainty that the referee gives the impression to seek is simply not possible with this data set. The plot "sigma_abs(ATN)" is only representative of an $ATN$ dependency if all $ATN$ intervals comprise the same number of data points. This is not the case in this data set as can be seen from the histogram of Fig. 8. Alternatively, one could pick data points at random from the data set to achieve a constant number of data points for a range of $ATN$ values. It would still be questionable if this would represent a true $ATN$ dependency or something else. The second best thing, in the authors' opinion, is to compare $\sigma_0(ATN)$ to $\sigma_{ap}$ which gives the comparison a point of reference; this is already shown in Fig. 8 of the manuscript. As shown in Appendix B, an $ATN$ dependency of $\sigma_{ap}$ is not expected to fundamentally change Fig. 8. Also added to Fig. 8 are the slopes of the ATN dependency for reference.

A figure was added showing $C_f$ values as a function of light scattering as requested by the referee (Fig. 9 in the revised manuscript). The relationship between $C_f$ is indeed dependent on the amount of light scattering by the aerosol. This is expected since apparent absorption has not been compensated for in $\sigma_0$. The new figure is discussed in two additional paragraphs in section 4.2.

**Comment:** In addition to this, the Aethalometers are compared to different instruments: two different versions of PSAP, CLAP and MAAP. The authors need to substantiate that the comparison of the Aethalometers to these different instruments is relevant. No comparison between the PSAPs, CLAP and MAAP is reported. The authors should at least sum up the results of laboratory inter-comparisons if no comparisons for ambient Arctic measurements is available.

**Reply:** A review of relevant studies was added to section 2.4 of the manuscript discussing relevant literature for this study. As no such intercomparison exists for the Arctic, the discussion relies mostly on laboratory experiences. In the lab experiments, also aerosols with high single-scattering albedo have been investigated, which is of relevance to this work. Urban measurement comparisons are not that relevant to this work as the filter artefacts are quite different for dark aerosol (Sheridan et al. 2005, Petzold et al. 2005).

**Specific comments**

**Comment:** Page 2, line 22: "below the detection limit". As the authors later point out, the detection limit is a function of the time between two consecutive measurements and the averaging time. One can lower the detection limit by integrating the sample for a longer time. The sentence needs to be modified or the time resolution (5 min?) needs to be specifically mentioned.

**Reply:** The detection limit of the instrument is only indirectly a function of the averaging time because the detection limit depends on the change in $ATN$ and the uncertainty in the $ATN$ measurements. Figure 1 demonstrates the difficulty of defining a detection limit as a function of averaging time because of drift. Moreover, the drift is instrument specific (Fig. 3). To elaborate on the detection limit being a function of $ATN$ and its relation to time and drift seems unnecessary at this point in the manuscript.

**Comment:** P. 3-5: it would be a good idea to report the inlet cuts (PM2.5. . .), flows (or face velocities), operational wavelengths of the filter photometers for all sites. The conditions for triggering the change of tape should also be reported (8 hours in Pallas, for example, elsewhere an ATN limit).

**Reply:** Inlets, flow rates, wavelengths, and filter change settings were added to the manuscript as a new table on page 3 (Before section 2.1 Measurement sites).

**Comment:** P. 5, l. 14: "Initially, when no aerosol particles have been deposited onto

the filter, light is transmitted through the filter with an intensity I_0." This is not true, the Aethalometers measure I_0 (intensity of light transmitted through the reference part of the filter without any sample) at the same time as the intensity of light I transmitted through the sample. Please change.

**Reply:** The sentence was changed to: "Light is transmitted through a pristine part of filter with an intensity $I_0$. The light that traverses through the part where aerosol particles deposit is transmitted with an intensity $I$."

**Comment:** P. 5, l. 20 and repeated later: "uncorrected light absorption coefficient ($\sigma 0$)". This is the "attenuation coefficient", please see above. Please change accordingly.

**Reply:** Changed to 'attenuation coefficient'.

**Comment:** P. 5, l. 31: "14625/$\lambda$ m2g–1". Please use unitless expressions, for example "16.6 m2g–1 at 880 nm, scaling inversely with the wavelength" or something similar.

**Reply:** The sentence was changed according to the referee's wishes. The new sentence now reads: The conversion from $\sigma_0$ to eBC is done using a wavelength-dependent mass attenuation cross-section (MAC$_{AE}$) and calculated using 16.62 m$^2$g$^{-1}$, scaling inversely with wavelength (e.g. Arnott et al., 2005).

**Comment:** P. 7. Section 2.4: the comparison of the so-called "reference" instruments needs to be elaborated (please see above). Has there been an intercomparison of the instruments in question? In ambient conditions or in the laboratory? How are published results of other intercomparisons relevant for the results reported in the manuscript?

**Reply:** An additional paragraph was added to the now rephrased Section '2.4 Co-located filter-based absorption instruments' on previous inter-comparisons of the instruments in question. The most relevant intercomparison is the GAW2005 and EU-SAARI2007 workshop paper by Müller et al. (2011). That paper is a laboratory intercomparison paper which includes, in addition to very dark aerosols, a substantial

section on non-absorbing aerosol. The second most relevant paper, by Petzold et al. (2005), which is also discussed in the revised version of the manuscript. No inter-comparisons between the instruments in question exist for the Arctic and most other intercomparisons, those not listed here, are for considerably darker aerosol. The impli-cations of these studies are discussed in the revised version of the manuscript.

**Comment:** P. 8., l. 9: is the 2% relative uncertainty for spot area specific to Aethalome-ters or to all filter photometers. Can the diameter be measured to 0.1 mm? How do relative and absolute uncertainties of the "reference" filter photometers influence the results?

**Reply:** In theory, the spot size area can be measured as accurately as the edge of the spot is sharp. In practice, using a magnifying glass, the diameter can be determined with roughly 0.1 mm accuracy. Using digital imaging techniques, however, the area can be determined with a 2% accuracy. This is done from a photograph of the filter and a scale which is then analysed digitally to determine the area.
The relative and absolute uncertainties of the reference instruments were not explicitly investigated here so the discussion is limited to previous research. The relative un-certainties of the co-located absorption photometers impact the results according to uncertainty propagation as

$$\frac{\delta C_f}{C_f} = \sqrt{\left(\frac{\delta\sigma_0}{\sigma_0}\right)^2 + \left(\frac{\delta\sigma_{ap}}{\sigma_{ap}}\right)^2} \quad (1)$$

The first term under the root is depicted in Fig. 4 and discussed extensively in the manuscript. This is the ideal case with no drift, only electronic noise. The second term under the root, namely $\delta\sigma_{ap}/\sigma_{ap}$ can be estimated from literature. The term $\delta\sigma_{ap}/\sigma_{ap}$ is generally considered to be in the order of 20–30% for the PSAP, and therefore also the CLAP (Bond et al. 2013; Sherman et al. 2015) and 12% for the MAAP (Petzold et al. 2005).

For boxcar averaging, which was used to match the collection time approach of the Aethalometers with the co-located filter-based absorption photometers, the absolute measurement uncertainty reduces proportionally to the averaging time ($t_{\mathrm{avg}}$) as $\delta\sigma_{\mathrm{ap}} \approx t_{\mathrm{avg}}^{-0.5}$ for the PSAP; and therefore also for the CLAP (Springston and Sedlacek, 2007). For the MAAP, boxcar averaging will also reduce the measurement uncertainty proportional to $t_{\mathrm{avg}}^{-0.5}$. The need for a longer time base, whether collection time or boxcar averaging, is derived from the Aethalometers. Since the instruments sample the same ambient air, the need for longer time bases will be the same. Consequently, prolonged Aethalometer collection times will improve the detection limit of the co-located instruments as well. This discussion was added to the revised manuscript.

**Comment:** P. 8, l. 19: "The one wavelength Aaethalometer at Summit was interpolated from 880 nm to 637 nm using a $\alpha$ of −1 in Eq. (5)." Please correct the typo "Aaethalometer" → "Aethalometer". What is the real value of the absorption Angstrom exponent $\alpha$? Could the lower C determined at Summit be (partly) due to the systematic bias due to the extrapolation from 880 nm to 635 nm using an $\alpha$ value that is too small? Are there any multi-wavelength measurements of absorption at Summit (sat least a short time series)?

**Reply:** The typo was corrected. The CLAP at Summit is a three-wavelength instrument. The average Ångström exponent as measured by the CLAP was 0.815. In the revised manuscript this exponent was used for the interpolation to 637 nm and the results were updated accordingly. Using the Ångström exponent of 0.815 changes the observed difference between Summit and the other stations slightly but not significantly. In fact, the lower $\alpha$ changes the median Cf value from 1.57 to 1.50 for Summit. While revisiting the code, a bug in the code was found which changes the decimals in Table 6. The table was updated accordingly.

**Comment:** P. 9, l. 22: "Moreover, lateral flow can influence both the signal and reference detectors, and thus ATN, through deposition of aerosol particles that do not originate from the sample air stream." This is highly unlikely, as the particles will get

filtered out at the edges of the filter material, not above the light detectors in the filter photometers. Please substantiate the claim or remove the sentence.

**Reply:** It is indeed highly unlikely. The sentence was therefore removed. Revisiting the sensing and reference detector data shows that the drop in $ATN$ during the lab zero air measurements is from the drop in intensity through the sample spot. The lateral flow is most likely to only influence the uncertainty in the flow rate and not the light transmission measurements themselves.

**Comment:** P. 10-11, Fig. 1: The sources of the drift in the laboratory experiment with the absolute filter are intriguing. The authors should at least offer some hypothesis on them and comment whether they are relevant for ambient measurements. How does the additional pressure drop due to the filter influence the measurements? Does it in fact introduce additional drift? Are there jumps or transients when tape is moved and measurements restarted? Do semi-volatile organic compounds adsorb on the filter and cause a signal, appearing as drift? Is the air conditioning in the laboratory important? Does drift have a wavelength dependence (indicating sample deposition rather than a fixed electronic drift)?

**Reply:** Three paragraphs were added following the introduction of Fig. 1 discussing the drift and its possible including the topics suggested. A figure was added to an Appendix showing the Aethalometer flow rate, room temperature, and $ATN$ for all wavelengths for the lab experiments. We can only hypothesise that semi-volatile organic compounds could adsorb on the filters. The adsorption would, however, be hampered by the absolute filter in front of the instrument that should adsorb the bulk of any semi-volatile organics in the air.

**Comment:** P. 11-12, Fig. 3: Was the absolute filter attached to the Aethalometer or to the sampling line leading to the Aethalometer? Were other instruments attached to the same sample line? Were there any pressure drops, jumps, transients in the sample line, or events in the measurement room, resulting in movement of the filter in the

Aethalometers? Any movement or drift in the filter position influences the measurement of ATN. The authors need to comment on these possible sources of drift and the observed transients.

**Reply:** Three paragraphs were added to the manuscript discussing the points raised by the referee. In addition, additional graphs were added the Appendix (Fig. A2) showing a more detailed picture of the measurements than what was possible to squeeze into Fig. 3.

**Comment:** P. 13, l. 32-33: "Thus, the drift uncertainty seen in Fig (3) becomes 0.003–0.03 Mm–1 after multiple scattering correction is applied." The effect of the reduction of noise on the uncertainty of C and the effect of thus derived C on the uncertainty of the absorption coefficient should be presented more clearly, starting perhaps here.

**Reply:** The reduction of noise will make the determination of $C_f$ less uncertain. When $\sigma_{ap}$ is low, the time window used for the calculations will be adapted to instrument response and therefore also lowered when noise is expected to be an issue. The determination of $C_f$ is affected by noise in the measurement as well as uncertainties in the measurement method itself. This is discussed in more detail in the revised version of the manuscript beginning with three paragraphs that were added to the beginning of section 4.2.

**Comment:** P. 14, Table 4: The determination of the C within ACTRIS "grey literature" reports would be a valuable reference here.

**Reply:** A reference was added to the table as the referee suggested.

**Comment:** P. 14, l. 22-25: "Third, it has to be acknowledged that there can be a bias in the absolute Cref values because of imperfect corrections of filter artefacts in the reference instruments (Backman et al., 2014; Müller et al., 2011). However, this bias should not substantially alter the ATN dependency because filter changes were not performed in sync." This is an oversimplification. At constant concentration and

equally spaced movements of tape (but not synchronized, and dependent on the flow of the individual instruments) an back-side-of-an-envelope calculation shows that the effect can be up to 20%. This is unlikely for global background sites in the Arctic, but this needs to be shown at least in the Supplement of the manuscript.

**Reply:** It is not clear how the referee conceived the value 20%. However, let us consiser a test case with a constant concentration of $\sigma_{\mathrm{ap}}$ = 2 Mm$^{-1}$. The Aethalometer $\sigma_0$ values have an $ATN$ dependency of $k$=0.002 (from Virkkula et al. 2015 for a high single-scattering albedo aerosol). The term $k$ comes from $\sigma_{\mathrm{ap}} = (1 + kATN)\sigma_0$ so that (ideally) $\sigma_{\mathrm{ap}}$ does not have an $ATN$ dependency. Solving $\sigma_0$ provides the equation for an $ATN$ dependent $\sigma_0$. For simplicity, the $ATN$ time series was calculated directly from $\sigma_{\mathrm{ap}}$ as $\Delta ATN = Q\sigma_0\Delta t/A$ so that the $ATN$ increase started from 0 after a threshold of 85 was reached. These calculations were repeated for two instruments with different flow rates and spot sizes to simulate non-synced filter changes. One instrument (Aethalometer) has a flow rate of 5 lpm, a spot size of 0.5 cm$^2$ and the co-located instrument (e.g. a PSAP) had a flow rate of 1.1 lpm, spot size of 20 mm$^2$; a $\Delta t$ of 60 min was used in the calculations. To simulate a non-perfect co-located instrument, a $k$ value of 0.001 was used to calculate an $ATN$ dependent $\sigma_{\mathrm{ap}}$ from a co-located instrument. That simulates that the correction algorithm used for the co-located instruments was only able to compensate for half of the filter induced artefacts arising from filter loading effects; or alternatively overcompensated the loading effects. The time series of the synthetic data is shown in the left panel in Fig. 1 below.

Using the $ATN$ dependent $\sigma$ap ($k$=0.001) representing a co-located PSAP to calculate $\sigma_{\mathrm{ap}}/\sigma_0$ and plotting it as a function of $ATN$ gives the right hand picture. The boxplot shows statistics of the $\sigma_{\mathrm{ap}}/\sigma_0$ ratio for different $ATN$ bins. Curve fitting to the median of each bin gives the $\sigma_{\mathrm{ap}}/\sigma_0$ ratio as a function of $ATN$ which is the equation $\sigma_{\mathrm{ap}}/\sigma_0 = 1 + k\,ATN$. The curve fit gives a $k$ value of the simulated Aethalometer of 0.0019 which is 5% less than expected (it should be 0.002). Varying the $k$ value of the co-located instrument between -0.004 and 0.004 yields $k$ values of the simulated Aethalometer that are between -27% and 38% from the expected $k$ values.

The $\sigma_{\mathsf{ap}}/\sigma_0$ ratio is equivalent to $C_{\mathsf{f}}^{-1}$ and should therefore also apply to Fig. 8. The exercise was added to the supplement material with the addition of a time series with a randomly changing $\sigma_{\mathsf{ap}}$. For the changing $\sigma_{\mathsf{ap}}$ time series, the results are nearly identical with the exception more data points are needed to lower the difference between the estimated and actual $k$ values.

**Comment:** P. 15, Figure 9: The regression equation on the figure is missing sigma_ap as the independent variable, making it confusing for the reader.

**Reply:** The mistakenly omitted $\sigma_{\mathsf{ap}}$ was added to the equation in the figure.

**References**

Bond, T. C., Anderson, T. L., and Campbell, D.: Calibration and intercomparison of filter-based measurements of visible light absorption by aerosols, Aerosol Sci. Technol., 30, 37–41, 1999. Collaud Coen, M., Weingartner, E., Apituley, A., Ceburnis, D.,

Fierz-Schmidhauser, R., Flentje, H., Henzing, J. S., Jennings, S. G., Moerman, M., Petzold, A., Schmid, O., and Baltensperger, U.: Minimizing light absorption measurement artifacts of the Aethalometer: evaluation of five correction algorithms, Atmos. Meas. Tech., 3(2), 457–474, doi:10.5194/amt-3-457-2010, 2010. Hyvärinen, A.-P., Vakkari,

V., Laakso, L., Hooda, R. K., Sharma, V. P., Panwar, T. S., Beukes, J. P., van Zyl, P. G., Josipovic, M., Garland, R. M., Andreae, M. O., Pöschl, U. and Petzold, a.: Correction for a measurement artifact of the Multi-Angle Absorption Photometer (MAAP) at high black carbon mass concentration levels, Atmos. Meas. Tech., 6(1), 81–90, doi:10.5194/amt-6-81-2013, 2013. Müller, T., Henzing, J. S., de Leeuw, G., Wiedensohler, A., Alastuey, A., Angelov, H., Bizjak, M., Collaud Coen, M., Engström, J. E., Gruening, C., Hillamo, R., Hoffer, A., Imre, K., Ivanow, P., Jennings, G., Sun, J. Y.,

[Figure]

Kalivitis, N., Karlsson, H., Komppula, M., Laj, P., Li, S.-M., Lunder, C., Marinoni, A., Martins dos Santos, S., Moerman, M., Nowak, A., Ogren, J. A., Petzold, A., Pichon, J. M., Rodriquez, S., Sharma, S., Sheridan, P. J., Teinilä, K., Tuch, T., Viana, M., Virkkula, A., Weingartner, E., Wilhelm, R., and Wang, Y. Q.: Characterization and intercomparison of aerosol absorption photometers: result of two intercomparison workshops, Atmos. Meas. Tech., 4, 245–268, doi:10.5194/amt-4-245-2011, 2011. Petzold, A.,

Schloesser, H., Sheridan, P. J., Arnott, W. P., Ogren, J. A., and Virkkula, A.: Evaluation of multiangle absorption photometry for measuring aerosol light absorption, Aerosol Sci. Technol., 39(1), 40–51, doi:10.1080/027868290901945, 2005. Sheridan, P. J.,

Arnott, W. P., Ogren, J. A., Andrews, E., Atkinson, D. B., Covert, D. S., Moosmüller, H., Petzold, A., Schmid, B., Strawa, A. W., Varma, R., and Virkkula, A.: The Reno aerosol optics study: An evaluation of aerosol absorption measurement methods, Aerosol Sci. Technol., 39(1), 1–16, doi:10.1080/027868290901891, 2005. Sherman, J. P., Sheridan, P. J., Ogren, J. A., Andrews, E., Hageman, D., Schmeisser, L., Jefferson, A., and Sharma, S.: A multi-year study of lower tropospheric aerosol variability and systematic relationships from four North American regions, Atmos. Chem. Phys., 15, 12487-12517, doi:10.5194/acp-15-12487-2015, 2015. Virkkula, A., Ahlquist, N. C., Covert, D. S., Arnott, W. P., Sheridan, P. J., Quinn, P. K., and Coffman, D. J.: Modification, calibration and a field test of an instrument for measuring light absorption by particles, Aerosol Sci. Technol., 39(1), 68–83, doi:10.1080/027868290901963, 2005. Virkkula, A., Chi,

X., Ding, A., Shen, Y., Nie, W., Qi, X., Zheng, L., Huang, X., Xie, Y., Wang, J., Petäjä, T., and Kulmala, M.: On the interpretation of the loading correction of the aethalometer, Atmos. Meas. Tech., 8, 4415-4427, doi:10.5194/amt-8-4415-2015, 2015.

[Figure]

**Fig. 1.** Simulated ATN dependency of a co-located instrument and an Aethalometer (left) and the resulting ATN dependency when the two are compared.